

# Ocean acidification enhances primary productivity and nocturnal carbonate dissolution in intertidal rock pools

**Narimane Dorey[1, †], Sophie Martin[2, 3], Lester Kwiatkowski[4]**

[1] LMD-IPSL, CNRS, Ecole Normale Supérieure/PSL Res. Univ, Ecole Polytechnique, Sorbonne Université, Paris, 75005, France

[2] CNRS, UMR7144, Station Biologique, Place Georges Teissier, 29688 Roscoff Cedex, France

[3] Laboratoire Adaptation et Diversité en Milieu Marin, Sorbonne Universités, UPMC Univ Paris 06, Station Biologique, Place Georges Teissier, 29688 Roscoff Cedex, France

[4] LOCEAN Laboratory, Sorbonne Université-CNRS-IRD-MNHN, Paris, 75005, France

[†] Correspondence to: Narimane Dorey

Ecole Normale Supérieure,

Département de Géosciences,

24 rue Lhomond, 75005 Paris, France

E-mail:    narimane.dorey@gmail.com





**ABSTRACT**
Human $CO_2$ emissions are modifying ocean carbonate chemistry, causing ocean acidification, and
likely already impacting marine ecosystems. In particular, there is concern that coastal, benthic
calcifying organisms will be negatively affected by ocean acidification, a hypothesis largely
supported by laboratory studies. The inter-relationships between carbonate chemistry and marine
calcifying communities *in situ* are complex and natural mesocosms such as tidal pools can provide
useful community-level insights. In this study, we manipulated the carbonate chemistry of
intertidal pools to investigate the influence of future ocean acidification on net community
production (NCP) and calcification (NCC) at emersion. Adding $CO_2$ at the start of the tidal
emersion to simulate future acidification ($+1500$ µatm $p$CO$_2$, target pH: 7.5) modified net
production and calcification rates in the pools. By day, pools were fertilized by the increased $CO_2$
($+20$ % increase in NCP, from 10 to 12 mmol $O_2$ m$^{-2}$ hr$^{-1}$), while there was no measurable impact
on NCC. During the night, pools experienced net community dissolution (NCC $< 0$), even in
present-day conditions, when waters were supersaturated with regards to aragonite. Adding $CO_2$
in the pools increased nocturnal dissolution rates by 40% (from -0.7 to -1.0 mmol CaCO$_3$ m$^{-2}$ hr$^{-1}$
) with no consistent impact on night community respiration. Our results suggest that ocean
acidification is likely to alter temperate intertidal community metabolism on sub-daily timescales,
enhancing both diurnal community production and nocturnal calcium carbonate dissolution.
**SHORT SUMMARY**
Human CO2 emissions are modifying ocean carbonate chemistry, causing ocean acidification, and
likely already impacting marine ecosystems. Here, we added $CO_2$ in intertidal pools at the start of
emersion to investigate the influence of future ocean acidification on net community production
(NCP) and calcification (NCC). By day, adding $CO_2$ fertilized the pools ($+20$ % NCP). By night,
pools experienced net community dissolution, a dissolution that was further increased ($+40$ %) by
the $CO_2$ addition.
**Keywords**: Ocean acidification, calcification, coralline algae, mesocosms, primary production,
temperate community, tidal pool



## INTRODUCTION

The ongoing increase of anthropogenic carbon dioxide ($CO_2$) in the atmosphere and the ocean – resulting in ocean acidification - is likely to create adverse living conditions for marine coastal communities (IPCC, 2019). Ocean acidification is projected to further decrease average surface pH by up to 0.4 units by 2100 (Kwiatkowski et al., 2020), and is identified as a major threat to marine ecosystems (IPCC, 2019). Lower seawater pH has significant effects on marine organisms physiology and fitness: from altered survival and reduced growth (see review by Kroeker et al. 2013), to changes in pH homeostasis (e.g., Kottmeier et al., 2022), metabolic rates, and energy trade-offs (e.g., Dorey et al., 2013; Pan et al., 2015) and reduced feeding efficiency (e.g., Stumpp et al., 2013). Marine calcifiers - the builders of calcified structures ($CaCO_3$) - have been a focus of ocean acidification research due to the sensitivity of calcification to the carbonate saturation state ($\Omega$), defined as follows:

$$\Omega = [Ca^{2+}]\ [CO_3^{2-}]/K'_{sp}$$

where $K'_{sp}$ is the stoichiometric solubility product for the considered carbonate polymorph (i.e., $\Omega_a$ for aragonite or $\Omega_c$ for calcite). The saturation state depends on temperature, pH, and pressure (lower $\Omega$ when pH or temperature decreases and pressure increases). When $\Omega < 1$, inert carbonate minerals tend to dissolve. The polymorphs composing the calcified structure like calcite and to a greater extent aragonite and high-magnesium calcite, are prone to dissolution when pH decreases. For instance, in Atlantic surface waters (at 20°C), saturation state equilibrium ($\Omega = 1$) is reached at pH 7.3 ($pCO_2$ = 2650 µatm) for calcite but at pH 7.6 (1250 µatm) for aragonite. For high-magnesium calcite, experiments from (Yamamoto et al., 2012) demonstrate that inert (dead) high-magnesium calcite from coralline algae passively dissolves at $\Omega_a$ values between 3.0 and 3.2 (also see Ries et al. 2016). Organisms with calcified structures are thus likely to experience reduced net calcification due to ocean acidification, both through enhanced dissolution, and reduced gross calcification rates.

Aside from acidifying the ocean (increased $H^+$), increased ocean $CO_2$ uptake could affect the productivity of algae and marine plants. As $CO_2$ dissolves in the ocean, the dissolved inorganic carbon (DIC: $CO_2$, $HCO_3^-$ and $CO_3^{2-}$) concentration increases. DIC is the substrate for marine photosynthesis (mainly $CO_2$ and $HCO_3^-$), and as such, it can limit photosynthetic rates when scarce.



In algae and marine plants that are carbon limited (permanently or periodically), elevated DIC could also directly increase photosynthetic rates and Mackey et al., (2015) propose that these rates could be further increased by the higher concentration gradient between water and the photosynthetic cells. However, the authors point out that while positive effects are theoretically expected, they may be small, specific to species' biology and the environment they live in, and difficult to predict (see also Hurd et al., 2019). In terrestrial ecosystems, the Intergovernmental Panel on Climate Change defines $CO_2$ fertilization as 'the enhancement of plant growth as a result of increased atmospheric $CO_2$ concentration' (Jia et al., 2019) and reports that $CO_2$ fertilization has likely already happened, although the magnitude of this effect depends on the plants, or assemblages/ecosystems considered (and on other factors constraining growth).

The response of single species to changes such as ocean acidification and increased DIC concentrations are often insufficient to predict community-level impacts. Ecological interactions such as competition or predation can affect the outcome of perturbation experiments (Kroeker et al., 2012). For instance, Paiva et al. (2021) showed that the laboratory growth of an isopod species was an order of magnitude slower than when raised in the presence of other species from its community. In another study, Legrand et al. (2019) showed that the presence of grazers increased coralline algal calcification (+50% in winter and +100% in summer), but when grazers were combined with ocean acidification, algal calcification decreased more than with acidification alone. Not taking into account such interactions can therefore result in poorly characterizing the effects of ocean acidification. Furthermore, while critical for a mechanistic understanding of the processes affecting marine biota, laboratory studies are seldom realistic. Typically performed in controlled, simplified, and stable conditions (e.g., with respect to temperature and food), laboratory studies can better assess the effect of pH alone (Widdicombe et al., 2010). However exposure to a stable pH (e.g., 7.6 vs. 8.0), fails to reflect the daily and seasonal variability observed in natural ecosystems, in particular coastal ones (Torres et al., 2021). Natural mesocosm perturbation experiments are thus essential tools to investigate future changes in variable and complex ecosystems, difficult to capture in the lab (Andersson et al., 2015; Barry et al., 2010).

Most *in situ* mesocosm experiments investigating the effect of ocean acidification have been conducted on planktonic communities, kept in large "bags" equilibrated to the desired pH (Riebesell et al., 2013). These studies demonstrate that adding $CO_2$ can significantly change the



organization of the plankton community (Spisla et al., 2021), and increase autotrophic biomass in high-nutrient conditions (Schulz et al., 2013). Due to the technical challenges, however, benthic calcifying communities are seldom manipulated this way *in situ* (Widdicombe et al., 2010). Two such manipulation experiments are the studies by Albright et al. (2016, 2018), where the authors used NaOH and $CO_2$ to reproduce pre-industrial and future pH conditions on a coral reef and found evidence that reef growth had been reduced by 7% over the industrial era and was likely to decline further. Other studies have investigated such community-level effects by either simulating "artificial", simpler, assemblages in laboratory setups (e.g., Cox et al., 2015; Pansch et al., 2016) or using phenomena such as natural $CO_2$ vents. For instance, in the vents of Ischia, as pH decreases, the presence of calcifying species declines (see review by Foo et al., 2018). Alternatively, Kwiatkowski et al. (2016) used locally-induced acidification due to respiration (no $CO_2$ addition) in tidal pools, a naturally closed system, and demonstrated that nighttime dissolution of these communities was positively correlated with $\Omega$. Here, we used tidal pools of the English Chanel as ephemeral mesocosms, where we modified carbonate chemistry conditions at the start of emersion through $CO_2$ addition.

Temperate rocky tidal pools - or rockpools - are highly dynamic systems that have been long studied by naturalists since they are easy to reach and their ecosystem structure generally resemble subtidal benthic communities (Ganning, 1971). Tidal pool organisms from the upper shore, well-adapted to pool conditions, form typical benthic communities: often low in diversity, they consist of a few characteristic macroalgal (e.g., *Ulva sp.*) and animal species (e.g., limpets). In winter, red macroalgae – including calcifying algae – often dominate the pools and while they do not disappear in summer, a bloom of soft green macroalgae is generally observed during the warm season. Temperature, salinity, oxygen, and pH in the pools are extremely variable, often far outside the seasonal range of nearby free-flowing seawater (Legrand et al., 2018; Morris & Taylor, 1983). Tidal pools generally emerge from the ocean twice a day in regions of semidiurnal tides with the duration dependent on shore location and the tidal coefficient. On short timescales tidal pools act as closed systems, with carbonate chemistry easily manipulated and temporal changes reflecting *in situ* community metabolism (no water mass transport and negligible air-sea gas exchange).

In the present study, we used tidal pools as natural mesocosms to investigate the effect of ocean acidification on communities dominated by calcifying red algae. We measured diurnal and



nocturnal net community calcification and production (or respiration) following $CO_2$ addition
across three seasons (winter, spring, and summer), to assess how tidal pool community metabolism
may respond to end of the 21st century high ocean acidification (pH 7.5).

### MATERIAL AND METHODS

**Field site**

The experiments were performed on a rocky intertidal shore characterized by granitic substrate on
the North coast of Brittany, France, between 2019 and 2021. The beach of Bloscon (48°43'30.0"N
3°58'10.5"W) is situated in Roscoff at the entrance of the Bay of Morlaix and has a hydrology
principally affected by the waters of the English Channel and to a lesser extent the Penzé and
Morlaix rivers (**Fig. 1**). This area is characterized by strong, oscillating, semidiurnal tides of up to
9 m. Temperatures are generally low in the deeper flowing water (from 9-10 °C in winter to 16-17
°C in summer), and salinity is close to that of the adjacent Atlantic (~35; see **Supp. Mat. Fig. S1**
for detailed temperature and salinity data from the two nearby SOMLIT monitoring stations
Estacade and Astan, a network described in Cocquempot et al., 2019).

**Tidal pool characterization**

For this study, we chose five tidal pools with high coverage in calcifying algae (≥ 30% of the pool
surface area). Both crustose (CCA) and articulated (branching) coralline algae (ACA) were
present. The field site has an eastern exposure, resulting in full morning sun and relatively early
shade in the evening. Foreshore locations of the pools resulted in daily emersion year-round
including during neap tides (mid-tide, approx. 5-6 m above chart datum). Pools emerged for 6-7 h
during low-tide periods. During that time, pools were completely separated from the adjacent open
water and their depths were effectively constant in winter (low-evaporation season), an indication
that there was no seawater leakage.
The volume of each of the five pools (from 16 to 39 L; **Fig. 2**) was estimated in April 2021
at the end of the emersion period just before high-tide flooding, by measuring salinity changes
when a known volume of freshwater was added and well mixed. To estimate the pools' initial
volumes, we also took into consideration the measured salinity changes throughout the emersion



period to estimate evaporative losses and combined this with the volume directly lost through
water sampling (see below). The pool projected area and the relative area covered by each type of
algae were estimated from aerial photographs, with a scale and analyzed using ImageJ (U. S.
National Institutes of Health, Bethesda, Maryland, USA, *https://imagej.nih.gov/ij*). Pool area
ranged from 0.3 to 0.6 m$^2$ (**Fig. 2**). The pools had slightly different community composition with
dominant calcifying red algae represented by *Lithophyllum incrustans* (CCA: 30 to 71 % of the
benthic cover) and *Ellisolandia elongata* (ACA: 0 to 6 % of the benthic cover). The remaining
pool area was either free of algal cover with only bare granitic rock visible or covered by soft
macroalgae. In summer (September 2020 and 2021), the pools also hosted the green algae *Ulva*
*sp.* and *Enteromorpha sp.* (2 to 44 % of the benthic cover: see **Supp. Mat. Pools: Fig. SP1-2,** for
results detailed by season) and, in Pool E, small single branches of the brown algae *Sargassum*
*muticum*, covering less than 0.5 % of the pool. We also noted the presence of diverse heterotrophs
such as anemones, sea sponges, small gobies, and shrimps. Calcifying invertebrates were
represented by four gastropod species: *Phorcus lineatus*, *Patella ulyssiponensis*, *Patella vulgata*
and *Gibbula pennanti*.
**Study design and seawater manipulation**
Fieldwork was conducted during the low-tide emersion periods, day and night. We refer to the
period from the beginning to the end of the pool emersion as a "low-tide emersion period" and to
each seasonal sampling period as a "field session" (**Table 1**). We sampled during three seasons:
winter (February 2020 and 2021), spring (April 2021), and summer (September 2020 and 2021).
During each field session, all the pools experienced both "future" (approximately year 2100 under
high emissions) and present-day ("present", non-manipulated control) initial carbonate chemistry
conditions. During each low-tide emersion period (n = 23), we randomly selected two or three
pools in which we decreased pH to 7.5 at the start of the emersion. The following low-tide
emersion period, this was reversed and pools that had been subject to present-day conditions in the
previous low-tide emersion period were subject to future conditions and *vice versa*. However, due
to diverse constraints, in two of the 23 emersion periods all the pools were left under present-day
conditions.
**Table 1:** **Sampling schedule**: The dates of each field session are presented. Pools were monitored
throughout multiple low-tide emersion periods (diurnal and nocturnal).





| Season | Dates | Low-tide emersion periods (N) | |
| --- | --- | --- | --- |
| | | **Diurnal** | **Nocturnal** |
| Winter | 14-17 February 2020 | 2 | 0 |
| | 9-19 February 2021 | 8 | 2 |
| Spring | 28-29 April 2021 | 2 | 0 |
| Summer | 2-11 September 2020 | 5 | 1 |
| | 6-9 September 2021 | 0 | 3 |


In this experiment, we compared "present" and "future" seawater carbonate chemistry
conditions. To simulate "future" carbonate chemistry conditions, we added small volumes of $CO_2$-
enriched seawater (total of ~100-200 mL) at the start of the emersion period in 50 mL increments
until the well-mixed pool water reached the desired pH levels (pH = 7.6, reached in less than 10
min.). $CO_2$-enriched seawater was prepared by super-saturating adjacent seawater in $CO_2$ using a
high-pressure $CO_2$ cylinder.

**Sampling and measurement of seawater parameters**

*Temperature, salinity, pH, oxygen, and ammonium*: From the start of the emersion period, we
measured five parameters periodically using HACH-Lange (Loveland, USA) probes: temperature,
$pH_T$ (IntelliCAL PHC101, accuracy: ± 0.02 pH units), salinity (conductivity probe IntelliCAL
CDC401, ± 0.1 units), oxygen concentration (optical sensor IntelliCAL LDO101, accuracy: ± 0.1
mg $L^{-1}$ for 0 to 8 mg $L^{-1}$ ± 0.2 mg $L^{-1}$ for greater than 8 mg $L^{-1}$, maximum 22 mg $L^{-1}$) and $NH_4^+$
concentration (ion selective electrode IntelliCAL ISENH4181, range: 0.018 - 9000 mg $L^{-1}$ $NH_4^+$-
N). Pools were well-mixed before any measurement to assure no influence of gradients forming in
the pools. The measurement frequency during the emersion periods was every 15-20 min during
the day (n = 1392) and reduced to once an hour at night (n = 159), when temperature, pH and light
variations were limited or absent. pH was calibrated on the total scale ($pH_T$) using TRIS (2-amino-
2-hydroxy-1,3-propanediol) and AMP (2-aminopyridine) buffer solution with a salinity of 35.0,
following the recommendations from (Dickson et al., 2007).
*Total alkalinity:* Discrete samples for total alkalinity (TA) analysis were collected hourly. The
average time between two samples was 1.0 ± 0.2 hours (n = 492, median = 1.0) during daytime





and 1.4 ± 0.9 hours (n = 135, median = 1.0) during nighttime. Seawater (150 mL) was filtered with
0.7 μm GF/F borosilicates filters directly after sampling. These samples were stored in a dark cool
box until the end of the tide (max. 7 h). Upon return to the lab, they were stored at 4°C in the dark
until they were either analyzed within the week or poisoned with 50 μL of saturated $HgCl_2$ (see
"*sample processing*"). TA was assessed potentiometrically using 50.0 ± 0.5 g of seawater and a
semi-automated titration system (0.1 M HCl, Titrino 848 plus by Metrohm, Switzerland; electrode
calibrated on the National Bureau of Standards scale). TA was determined using Gran titration
(Gran, 1952) according to the method of Haraldsson et al. (1997) and verified against reference
standards provided by A. Dickson (Scripps Institute of Oceanography, University of South
California, San Diego, United States). TA samples were analyzed with single (n= 312) or duplicate
(n = 320) measurements (the median of the standard deviation between duplicates was 1.05 μmol
$kg^{-1}$). TA was salinity-normalized before further calculations, to take into account possible dilution
from rain or concentration from evaporation.
To take into account the influence of the changes in nutrients ($NO_3^-$, $NO_2^-$, $PO_4^{3-}$ and $NH_4^+$)
on the changes in TA (Gazeau et al., 2015), we sampled seawater for nutrients in winter (February
2020) and summer (September 2020). Samples were taken during the day at the start and end of
the emersion periods in the five pools. Around 60 mL of seawater was immediately filtered on 0.2
μm cellulose filters, stored in 125 mL polyethylene bottles in a cool dark box (max. 7h), and then
frozen at -20 °C until analysis. Nutrient concentrations were obtained using an AA3 auto-analyzer
(Seal Analytical) using the method from Aminot & Kérouel (2007). Changes in nutrient
concentrations were near-negligible contributions to TA changes throughout a low-tide emersion
period (< 6 μmol $kg^{-1}$ i.e., < 2% of the observed change in TA, see full details in **Supp. Mat.**
**Nutrients**) and thus are ignored here.
*Light measurements:* Surface irradiance (photosynthetically active radiation, PAR) was
continuously recorded (every minute) during experiments at the field station, using a Li-Cor flat
quantum light sensor (LI-190R) and logger (LI-1500, LI-COR, Germany).
*Adjacent waters:* Temperature, salinity, pH and TA (n = 5) were similarly sampled and measured
at the sampling site during ebb tide, for the three seasons.
**Carbonate chemistry calculations**





The carbonate system parameters (e.g., $pCO_2$, DIC concentration, $CO_3^{2-}$ concentration, and $\Omega_a$, the
aragonite saturation state) were calculated from the measurements of $pH_T$, TA, temperature and
salinity using the *R* package *seacarb* (Gattuso et al., 2021) with the default dissociation constants
recommended by Dickson et al. (2007), except for the low temperatures encountered in February
2021 where the refined constants of Sulpis et al. (2020) were used. When salinity decreased by
more than 1.5 units per hour, data were excluded to avoid rain effects in the present study. When
calculated DIC and $\Omega_a$ were negative, likely due to inaccuracies in the measurement and
computation of the carbonate system, values were approximated to be 0 (7/627 values).
**Biological activity calculations**
The rates of Net Community Calcification (NCC; mmol $CaCO_3$ m$^{-2}$ h$^{-1}$) and Net
Community Production (NCP) or Community Respiration (CR; mmol $O_2$ m$^{-2}$ h$^{-1}$ or mmol C m$^{-2}$
h$^{-1}$) were calculated between two consecutive sampling times. These rates respectively represent
the measured changes of net $CaCO_3$ precipitation and net organic carbon production (or oxygen
consumption) by the community. Positive NCC represents net $CaCO_3$ precipitation (gross
precipitation > dissolution) and negative rates represent net dissolution (dissolution >
precipitation). NCP is positive when the community primary production exceeds respiration and
negative when community primary production is less than respiration. We use CR (not NCP) for
nights, when there is no primary production (oxygen consumption and carbon release only).
NCC was calculated using the alkalinity anomaly method (Smith & Key, 1975). Briefly,
for each mol of $CaCO_3$ precipitated, two moles of $HCO_3^-$ combine with $Ca^{2+}$, and TA decreases
by two moles (*Eq. 1*). Two independent estimates of NCP (or CR) were calculated, one derived
from changes in $\Delta O_2$ (NCP$_{O2}$ or CR$_{O2}$) and one derived from $\Delta$DIC and NCC (NCP$_{DIC}$ or CR$_{DIC}$).
NCC and NCP (or CR) were thus calculated as follows:
$$\text{NCC} = \frac{\Delta TA}{2\Delta t} \times \frac{V}{S} \qquad (1)$$
$$NCP\ (or\ CR)_{O2} = \frac{\Delta O_2}{\Delta t} \times \frac{V}{S} \qquad (2)$$
$$NCP\ (or\ CR)_{DIC} = \frac{-\Delta DIC}{\Delta t} \times \frac{V}{S} - NCC \qquad (3)$$



with $\Delta$TA (mmol L$^{-1}$), $\Delta$DIC (mmol L$^{-1}$) and $\Delta$O$_2$ (mmol L$^{-1}$) the change in concentration of TA,
DIC and O$_2$, between consecutive samples and $\Delta$t the duration between consecutive samples (h),
V pool volume (L), S the pool surface area (m$^2$).
Up to seven NCC and NCP (or CR) rates were calculated for each pool during each
emersion period (one per hour). These rates were used to investigate the direct correlation between
biological activity and environmental factors such as light intensity or $\Omega_a$.
Rates calculated this way are however not independent from each other (i.e., the rate
measured at t+2 is dependent on the rate at t+1), limiting further statistical analyses on the effect
of the treatment. This is why, to investigate the effect of pH treatment ("present" vs. "future") on
community biological activity, we also calculated NCC and NCP or CR using linear regressions
(NCC$_{lm}$ and NCP$_{lm}$ or CR$_{lm}$) between TA, [DIC] and [O$_2$] and time after the start of the emersion
period (for detailed results of the regressions, e.g., goodness-of-fit, see **Supp. Mat. LM1-3**). The
few data from diurnal tides that were taken after sunset were excluded from these regressions. For
oxygen, data was limited to the first three hours of emersion as high O$_2$ concentrations (>22 mg L$^{-1}$
$^1$) and supersaturation (>200 %) led to inaccurate measurements and/or possible oxygen degassing
afterwards (see **Supp. Mat. LM2**). This regression approach provides a single estimate of the rate
of NCC, NCP$_{DIC}$ (or CR$_{DIC}$) and NCP$_{O2}$ (or CR$_{O2}$) for each pool during each emersion period (n =
17 diurnal and 6 nocturnal low-tide emersion periods x 5 pools = 115). These rates were then used
in generalized linear mixed models (GLMM) to assess the effect of pH treatment on diurnal and
nocturnal biological activity (see "*statistical analyses*" below).
We calculated community calcification and production budgets (respectively CCB and
CPB) at emersion as an indication of the night/day balance in calcification and production: when
CCB/CPB is positive the pool community calcifies/produces more by day than they
dissolve/respire at night. Both were calculated for winter (February 2020 and 2021) and summer
(September 2020 and 2021) for each pool as follows:
$$CCB \; = NCC_{\,D} + \; NCC_{\,N} \qquad (4)$$
$$CPB \; = NCP_{\,D} + \; CR_{\,N} \qquad (5)$$



with $NCC_D$ and $NCP_D$ ($> 0$) the average diurnal NCC and NCP for a given pool for a treatment
and a season and $NCC_N$ and $CR_N$ ($< 0$) the average nocturnal NCC and dark respiration for the
same conditions. Three approaches were used for estimating CPB, given the uncertainties of each
NCP estimate (see discussion): (1) $O_2$-derived estimates of NCP ($CPB_{O2}$), (2) DIC- derived
estimates ($CPB_{DIC}$) and (3) a "mixed" approach that combined nocturnal $CR_{O2}$ and diurnal $NCP_{DIC}$
($CPB_m$), under the assumption that one mol of carbon is produced/consumed when one mol of $O_2$
is produced/consumed. Although CPB resemble gross community production in the way the rates
are calculated (difference between light and dark net production/respiration rates), if one wanted
to reuse these rates for gross community production, they should be do so with care due to
differences in night and day temperature (see extended discussion on this subject in Bracken et al.,
2022). The treatment effect was assessed on CCB and CPB by comparing the change due to the
"future" treatment in each pool.

**Statistical analyses**
All data are presented as mean $\pm$ standard deviation (SD). The analyses were made using the
software R (R Core Team, 2017). The level of significance used was 5%. Because data were
measured on the same five pools but on different days for different treatments, we used GLMM to
test for the effect of treatment on $NCC_{lm}$ and on $O_2$ and DIC-derived $NCP_{lm}$ (or $CR_{lm}$), assigning
sampling days (i.e., low-tide emersion periods) as the random factor and pools (five levels), mean
temperature of the pool during low-tide emersion period (a continuous proxy for season) and
treatment (*Treat*: "future" vs. "present") as fixed factors. This was performed using the *R* package
*nlme* (Pinheiro et al., 2018). Models with and without standardized residuals were compared using
ANOVAs and, when different, Akaike Information Criteria (AIC) was used to choose the best
fitted-model of the two. For GLMM, mean daily PAR was not used as it has strong collinearity
with mean daily temperature/season. We used ANOVAs to test the effect of temperature, pool and
treatment on initial (averaged over the first hour of emersion) and final (averaged for $> 5$ hours
after emersion) carbonate chemistry conditions. The normality of the data was tested using
Shapiro–Wilk tests and qq-plots, while variance homogeneity was tested with Bartlett tests.



## RESULTS

### 1/ Environmental conditions

**Adjacent waters:** Temperatures (and salinity) measured in the seawater adjacent to the pools were 6-7°C in winter (February; salinity S=35.0), 11-12°C in spring (April; S=35.5) and 17-18°C in summer (September; S=36.0). This seawater was characterized by average $pH_T$ of $8.01 \pm 0.06$ units, total alkalinity of $2319 \pm 6$ µmol kg$^{-1}$, $pCO_2$ of $445 \pm 69$ µatm, $\Omega_a = 2.2 \pm 0.3$ and $[O_2] = 100 \pm 1$ % of air saturation (or $10.1 \pm 1.5$ mg L$^{-1}$; n=5).

**Light duration and intensity:** In Roscoff, day:night periods are typically 10h:14h in February, 14h:10h in April and September. Photosynthetically active radiation (PAR) was two to three times higher in spring/summer (**Fig. 3A**: April/September ~1500 µmol m$^{-2}$ s$^{-1}$) than in winter (February ~500 µmol m$^{-2}$ s$^{-1}$).

**Carbonate chemistry conditions at the start of the emersion period (< 1h post emersion):** Both for diurnal and nocturnal tides, the initial pH was significantly lower in pools with added $CO_2$ than in the present-day pools (Day: $pH_T = 8.2 \pm 0.1$ vs. $7.5 \pm 0.2$ units; Night: $8.0 \pm 0.1$ vs. $7.4 \pm 0.1$ units for "present" and "future" pools respectively; *Treat p* < 0.001; detailed results in **Fig. S2-3**, **Table S1-2**). This corresponds to $pCO_2$ of $260 \pm 100$ vs. $1900 \pm 835$ µatm (day) and $510 \pm 90$ vs. $2310 \pm 410$ µatm (night) for pools in "present" and "future" conditions respectively. Adding $CO_2$ in the pools increased the mean DIC concentration by 320 µmol kg$^{-1}$ during the day and 240 µmol kg$^{-1}$ during the night. In "present-day" conditions, the pools started at supersaturated levels with regards to aragonite (day: $\Omega_a = 3.3 \pm 1.3$, night: $2.2 \pm 0.3$). Adding $CO_2$ significantly decreased $\Omega_a$ (Treat: *p* < 0.001, **Table S1**) leading to initial "future" conditions often undersaturated with regards to aragonite ($\Omega_a = 0.8 \pm 0.5$) by day and always undersaturated conditions by night ($\Omega_a = 0.6 \pm 0.1$). Furthermore, in "future" diurnal conditions, pools were always undersaturated with respect to aragonite from the start of the emersion period in February ($\Omega_a = 0.5 \pm 0.2$) but not in April ($\Omega_a = 1.1 \pm 0.7$) and September ($\Omega_a = 1.2 \pm 0.5$; **Table S1**). At the start of emersion, total alkalinity was $2303 \pm 34$ µmol kg$^{-1}$ (similar to adjacent seawater), and uncorrelated with treatment (*p* > 0.05) and temperature (*p* > 0.6).

As data was averaged on the first hour post-emersion, the mean initial oxygen concentration calculated was already affected by NCP by day ($14.0 \pm 2$ mg $O_2$ L$^{-1}$) and CR by





night ($9.5 \pm 1.5$ mg $O_2$ $L^{-1}$; vs. $10.1 \pm 1.5$ mg $O_2$ $L^{-1}$ for adjacent seawater). This was also visible
in $CO_2$ partial pressure, with lower $pCO_2$ than expected during the first hour post-emersion by day
($262 \pm 102$ µatm vs. $445 \pm 69$ µatm for adjacent seawater) and higher $pCO_2$ at night ($508 \pm 88$
µatm) in the "present-day" conditions.
**2/ Diurnal tides**
**Diurnal pool chemistry:** Starting from the aforementioned values at emersion, the pools followed
a clear temporal evolution due to solar irradiance and community metabolism (**Fig. 3**). Firstly, we
observed increases in salinity (+1.5 units on average, **Fig. 3A**) and temperature (+4°C in
September, +6°C in April on average) in summer and spring. In winter, temperatures tended to
decrease (-1.7°C on average) with air temperatures colder than that of the seawater; salinity was
stable ($35.5 \pm 0.8$).
Secondly, we observed positive NCP corroborated by a doubling in oxygen concentration
(**Fig. 3A**) a few hours after the start of emersion. In parallel, the seawater DIC concentration
decreased by half from the initial concentration (from $2130 \pm 195$ to $1140 \pm 560$ µmol $kg^{-1}$; **Fig.**
**3B**), the range of which largely depended on the season (**Fig. S2**). For instance, in February, DIC
consumption in pool seawater averaged ~700 µmol $kg^{-1}$ over a low-tide period, while it averaged
~1500 µmol $kg^{-1}$ in September. Particularly extreme conditions, with DIC concentrations
effectively reaching 0 µmol $kg^{-1}$, were observed in two of the pools, at three tides in September
2020 (see further details below in "*5/ The particular case of September 2020 tides*"). At the end
of diurnal emersions, average $pCO_2$ was always below 100 µatm, reaching as low as $1 \pm 2$ µatm
in September (**Fig. 3B**, **Table S1**). As a result, diurnal $pH_T$ increased to $9.1 \pm 0.6$ by the end of
emersion, with maximum values up to 10.3 in summer (**Fig. 3B**). At the end of a diurnal emersion
period, the pools' pH was stable, reaching either a plateau or decreasing after sunset (see PAR in
**Fig. 3A**). Similarly, at the end of diurnal emersion periods, $\Omega_a$ was high ($5.6 \pm 3.0$ on average;
max 10.4). Lastly, we observed a diurnal decrease in TA by 415 µmol $kg^{-1}$ on average, indicative
of net calcification.
It is noteworthy that the carbonate chemistry conditions experienced at the end of diurnal
emersion converged whatever the initial treatment (**Fig. S2**, **Table S1**). For instance, while $\Omega_a$ was
significantly different between treatments at the start of the emersion period, both treatments



384 reached similar $\Omega_a$ at the end of emersion (> 5 h) of around $5.3 \pm 2.2$ (ANOVA: Treat: $p = 0.1$,

385 Temp: $p = 0.002$, Pool: $p = 0.01$). There was less convergence for $pH_T$ where, even five hours after

386 emersion, there were still statistically significant, albeit small, differences between treatments ($p$

387 $< 0.001$ for $pH_T$ with $9.2 \pm 0.6$ for "present" and $9.0 \pm 0.6$ for "future" pools).

388 **Diurnal biological activity:** Net community production was positive during daytime, except at

389 sunset (**Fig. 3C**). NCP was significantly correlated to light intensity (PAR) and further results for

390 hourly NCP and their correlation to hourly averaged PAR, $\Omega_a$ and temperature can be found in the

391 **Supp. Mat.** (**Fig. S5** and **S6**).

392  As expected, seasons/temperature affected net oxygen production ($O_2$-derived $NCP_{lm}$),

393 increasing from $7 \pm 3$ mmol $O_2$ m$^{-2}$ hr$^{-1}$ in February to $18 \pm 11$ mmol $O_2$ m$^{-2}$ hr$^{-1}$ in September

394 (**Fig. 4A** and **Table 2A**: GLMM, $p < 0.001$). $CO_2$ addition increased $O_2$-derived $NCP_{lm}$ by 20%,

395 from $10 \pm 7$ mmol $O_2$ m$^{-2}$ hr$^{-1}$ in "present" conditions to $12 \pm 9$ mmol $O_2$ m$^{-2}$ hr$^{-1}$ ($p = 0.0015$). Net

396 oxygen production differed across pools ($p < 0.003$), with significantly more productivity in pool

397 C ($17.6 \pm 12.7$ mmol $O_2$ m$^{-2}$ hr$^{-1}$) and D ($10.6 \pm 5.6$ mmol $O_2$ m$^{-2}$ hr$^{-1}$), compared to the pools A,

398 B and E ($8.1 \pm 4.1$ mmol $O_2$ m$^{-2}$ hr$^{-1}$).

399  Results are similar for DIC-derived $NCP_{lm}$ (**Fig. 4A** and **Table 2B**), with primary

400 production ranging from $6 \pm 2$ mmol C m$^{-2}$ hr$^{-1}$ in February up to $12 \pm 5$ mmol C m$^{-2}$ hr$^{-1}$ in

401 September ($p < 0.001$). As for $O_2$-derived $NCP_{lm}$ $CO_2$ addition increased DIC-derived $NCP_{lm}$ by

402 20 % ($p < 0.001$, **Fig. 4A**). This increase was particularly apparent in the summer, where $NCP_{lm}$

403 increased from $11 \pm 4$ mmol m$^{-2}$ hr$^{-1}$ in the "present" treatment to $15 \pm 5$ mmol C m$^{-2}$ hr$^{-1}$ in the

404 "future" treatment (+ 35 %). Compared to pool A, productivity was significantly lower in pools B

405 and E and significantly higher in pools C and D ($p < 0.003$).

406  By day, with the exception of sunset, net community calcification was positive (NCC and

407 $NCC_{lm} > 0$: **Fig. 3C** and **4B**) and occurred in an environment that was supersaturated with regards

408 to aragonite (**Fig. 3B**). This was with the exception of a few emersion periods in September 2020

409 where dissolution was observed despite high saturation state conditions (further details below).

410 Similar to $NCP_{lm}$, diurnal net calcification rates ($NCC_{lm}$) were strongly influenced by

411 temperature/season (**Fig. 4B** and **Table 2C**: GLMM, $p < 0.001$) ranging from $1.2 \pm 0.5$ mmol

412 $CaCO_3$ m$^{-2}$ hr$^{-1}$ in February to $3.3 \pm 1.3$ mmol $CaCO_3$ m$^{-2}$ hr$^{-1}$ in September. NCC hourly rates




positively correlated with averaged $\Omega_a$ ($p < 0.0001$; NCC = 0.15 x $\Omega_a$ + 0.85; linear regression
presented in **Fig. S6**), significantly but not strongly ($R^2 = 10\%$). $CO_2$ addition did not influence
$NCC_{lm}$ rates during the day ($p = 0.47$). However, $NCC_{lm}$ did differ across pools ($p < 0.003$): rates
were relatively low in pool E – lowest CCA cover (30%) – (1.4 ± 1.4 mmol $CaCO_3$ m$^{-2}$ hr$^{-1}$), and
high in pool D – highest CCA cover (70%) – (2.2 ± 0.8 mmol $CaCO_3$ m$^{-2}$ hr$^{-1}$) compared to the
three other pools (2.0 ± 1.25 mmol $CaCO_3$ m$^{-2}$ hr$^{-1}$).
**Table 2: Results of the generalized linear mixed-effect models for A) O$_2$-derived NCP$_{lm}$**
**(mmol O$_2$ m$^{-2}$ hr$^{-1}$), B) DIC-derived NCP$_{lm}$ (mmol C m$^{-2}$ hr$^{-1}$) and C) NCC$_{lm}$ (mmol CaCO$_3$ m$^{-2}$**
**hr$^{-1}$) during the day and night**. The models include three fixed factors: *Temp* (mean temperature:
a continuous factor), *Treat* (for $CO_2$ "future" treatment vs. "present", two levels) and *pools* (vs. A,
five levels), and one random effect (*low-tide emersion period* or the calendar day at which the pool
was measured). Significant *p*-values are highlighted in bold.

| A. | O$_2$-derived NCP$_{lm}$ | | Estimate | Standard Error | *p*-value |
|---|---|---|---|---|---|
| Day | **Intercept** | | 1.49 | 1.37 | 0.28 |
| | **Fixed Effects** | *Temp* | 0.54 | 0.13 | **<0.001*** |
| | | *Treat* | 1.09 | 0.33 | **0.0015*** |
| | | *Pools* | A, B, E ≠ C, D | | **<0.003*** |
| | **Random Effect** | *Low-tide emersion period* | 8.90 | 0.90 | **<0.001*** |
| Night | **Intercept** | | 2.87 | 0.98 | **0.005*** |
| | **Fixed Effects** | *Temp* | -0.43 | 0.06 | **<0.001*** |
| | | *Treat* | -0.25 | 0.28 | 0.39 |
| | | *Pools* | A, B, D ≠ C, E | | **<0.03*** |
| | **Random Effect** | *Low-tide emersion period* | -3.46 | 0.87 | **<0.001*** |







| B. | DIC-derived NCP$_{lm}$ | | Estimate | Standard Error | *p*-value |
|---|---|---|---|---|---|
| **Day** | **Intercept** | | 2.3 | 1.08 | **0.035*** |
| | **Fixed Effects** | *Temp* | 0.38 | 0.08 | **<0.001*** |
| | | *Treat* | 1.25 | 0.25 | **<0.001*** |
| | | *Pools* | A ≠ B, C, D, E | | **<0.003*** |
| | **Random Effect** | *Low-tide emersion period* | 7.7 | 0.7 | **<0.001*** |
| **Night** | **Intercept** | | -0.94 | 1.4 | 0.51 |
| | **Fixed Effects** | *Temp* | -0.92 | 0.19 | 0.053 |
| | | *Treat* | -0.25 | 0.28 | **<0.001*** |
| | | *Pools* | A, B, D, E ≠ C | | **0.016*** |
| | **Random Effect** | *Low-tide emersion period* | 1.61 | 0.57 | **0.01*** |


| C. | NCC$_{lm}$ | | Estimate | Standard Error | *p*-value |
|---|---|---|---|---|---|
| **Day** | **Intercept** | | -0.16 | 0.31 | 0.61 |
| | **Fixed Effects** | *Temp* | -0.13 | 0.02 | **<0.001*** |
| | | *Treat* | 0.06 | 0.08 | 0.47 |
| | | *Pools* | A, B, C ≠ D, E | | **<0.003*** |
| | **Random Effect** | *Low-tide emersion period* | -1.90 | 0.24 | **<0.001*** |
| **Night** | **Intercept** | | 0.64 | 0.26 | **0.026*** |
| | **Fixed Effects** | *Temp* | 0.009 | 0.016 | 0.57 |
| | | *Treat* | 0.28 | 0.07 | **0.0017*** |
| | | *Pools* | A, B, C ≠ D, E | | **<0.017*** |
| | **Random Effect** | *Low-tide emersion period* | 0.83 | 0.078 | **<0.001*** |






### 3/ Nocturnal tides

**Nocturnal pool chemistry:** Seawater temperatures during the nights were stable (**Fig. 3A**) throughout the emersion period in summer (from $17.3 \pm 0.4$°C < 1 h post-emersion to $17.2 \pm 0.2$°C > 5 h post-emersion) and winter (from $8.4 \pm 1.4$°C to $7.8 \pm 2.7$°C in February; no April nights). We highlight the wide range of winter seawater temperatures with an exceptionally cold tidal cycle (5°C on the 13th of February 2021) due to air temperatures of 3-4°C (observations from the Île de Batz meteorological station). There was a decline in salinity at night in some winter emersion periods (**Fig. 3A**), due to high air humidity and/or rain. Data where salinity dropped by more than 1.5 units in less than an hour were removed from further analyses on net community calcification and respiration.

After five hours of emersion, $O_2$ concentration had decreased by half (from ≈10 mg $O_2$ $L^{-1}$ to $4.9 \pm 3.3$ mg $O_2$ $L^{-1}$) (**Fig. 3A**) due to community respiration. Simultaneously, $pH_T$ decreased to $7.6 \pm 0.2$ ("present") or stayed at $7.4 \pm 0.2$ ("future"; **Fig. 3B and S2, Table S1**), with significant effects of pools, treatment and temperature ($p < 0.001$ for all three). DIC concentration increased by +256 μmol $kg^{-1}$ on average over an emersion period. The range of this increase depended on the temperature and the pool: in winter (5-10°C), present-day pool seawater gained +130 μmol $kg^{-1}$ (+60 for "future" pools) of DIC over an emersion period, when in summer they gained +370 μmol $kg^{-1}$ for "present" ("future": +310 μmol $kg^{-1}$) pools. Saturation state converged towards similar undersaturated levels at night (**Fig. 3B and S2**, **Table S1**): $\Omega_a$ stayed stable in the "future" treatment ($0.7 \pm 0.2$ units on average) and decreased in the "present-day" treatment (-1.2 units from initial $\Omega_a$). At the end of nocturnal emersion $\Omega_a$ were still statistically different due to the initial treatment ($p < 0.001$ for *Treat*, *Temp* and *Pools*).

**Nocturnal biological activity:** At night, oxygen was consumed, i.e., we observed dark respiration (CR; **Fig. 3C**). Community respiration ($O_2$-derived $CR_{lm}$) varied according to season (**Fig. 4A** and **Table 2A**: $p < 0.001$): temperature linearly increased nocturnal respiration rates from $-1.0 \pm 1.2$ mmol $O_2$ $m^{-2}$ $hr^{-1}$ in February to $-4.7 \pm 1.3$ mmol $O_2$ $m^{-2}$ $hr^{-1}$ in September. The $CO_2$ treatment did not influence night respiration ($p = 0.39$). Respiration rates were significantly influenced by pools ($p = 0.03$), probably linked to the relative biomass of heterotrophs and autotrophs; respiration was significantly higher in pool C ($-4.6 \pm 2.8$ mmol $O_2$ $m^{-2}$ $hr^{-1}$) and significantly lower in pool E ($-2.4 \pm 1.4$ mmol $O_2$ $m^{-2}$ $hr^{-1}$) than in pools A, B and D ($-3.4 \pm 2.1$ mmol $O_2$ $m^{-2}$ $hr^{-1}$).



Night respiration estimated using DIC and NCC was near zero ($CR_{lm}$ = -0.2 ± 0.7 mmol m$^-$

$^2$ hr$^{-1}$). At these low rates, uncertainties associated with much higher rates of net dissolution
(negative NCC) sometimes led to spuriously positive DIC-derived CR estimates, hindering
interpretation. Nevertheless, DIC-derived community respiration was ten times lower in February
than in September (-0.2 ± 0.7 and -2.3 ± 1.1 mmol C m$^{-2}$ hr$^{-1}$ respectively), although it was not
linearly driven by temperature ($p$ = 0.053; **Fig. 4B** and **Table 2B**). Adding $CO_2$ to the pools
influenced DIC-derived community respiration in a way that was inverse to that seen with $O_2$, but
as stated above, this was likely an artifact of subtracting NCC from small DIC changes. As for $O_2$,
DIC-derived $CR_{lm}$ significantly changed depending on the pools.

At night, the pools experienced significant net community dissolution (NCC < 0: **Fig. 3C**)

even when waters were supersaturated with regards to aragonite in the "present" treatment (**Fig.**
**3B**: $\Omega_a$ > 1). Nocturnal net dissolution rates ($NCC_{lm}$) were not significantly affected by temperature
in the range investigated (5-18°C; **Fig. 4C** and **Table 2C**: $p$ = 0.57). However, adding $CO_2$ in the
pools increased net dissolution rates ($p$ = 0.0017) from -0.7 ± 0.3 mmol CaCO$_3$ m$^{-2}$ hr$^{-1}$ to -1.0 ±
0.4 mmol CaCO$_3$ m$^{-2}$ hr$^{-1}$ (+40 %). Similarly, looking instead at hourly rates (NCC), dissolution
correlated significantly ($p$ < 0.0001) with $\Omega_a$ (NCC = 0.34 x $\Omega_a$ – 1.22; $R^2$ = 11 %; **Fig. S6**). The
strength of this correlation depended on seasons and pools (**Fig. S7**). Net dissolution rates ($NCC_{lm}$)
significantly differed by pool ($p$ < 0.0017): the lowest rates were observed in pool E (-0.4 ± 0.2
mmol CaCO$_3$ m$^{-2}$ hr$^{-1}$) – the pool with the lowest CCA cover –, and the highest dissolution in pool
D – the pool with the highest CCA cover (-1.0 ± 0.4 vs. -0.9 ± 0.3 mmol CaCO$_3$ m$^{-2}$ hr$^{-1}$ for A, B
and C).

**4/ Influence of the treatment on CPB and CCB**
Pools fixed more carbon during the day than they respired at night, i.e., the community production
budget (CPB: balance between night and day) was positive in all the pools, both in winter and
summer and whatever the treatment (**Fig. 5**). $CPB_{DIC}$ and $CPB_m$ estimates were typically lower
than $CPB_{O2}$ (in 14/20 cases and 18/20 cases respectively). The production budget was significantly
lower in winter than in summer (FEB: $CPB_{O2}$ = 3 ± 1 mmol O$_2$ m$^{-2}$ h$^{-1}$, SEP: 7 ± 5 mmol O$_2$ m$^{-2}$
h$^{-1}$; t-test: t = -2.4, df = 9.8, $p$ = 0.03). Adding $CO_2$ increased CPB in all the pools in summer by +
3.0 ± 2.1 mmol O$_2$ m$^{-2}$ h$^{-1}$, an increase in production by 50 to 80 % (ΔCPB; **Fig. 5**). In winter, there



was no evidence of such a "fertilization effect" across the most accurate CPB estimates for this
season ($CPB_{O2}$, $CPB_m$): we only observed a significant increase in production due to $CO_2$ addition
in two of the pools (+60 % to +120 % for A and B). For the three other pools, CPB either induced
minimal changes (< 20 % for C and E) or a decrease in production (D: down to -34 %). DIC-
derived ΔCPB in winter (all positive) should be interpreted with caution since some nocturnal
$CR_{lm}$ were spuriously positive in the "future" treatment (see "*nocturnal biological activity*" above).

The pools calcified more during the day than they dissolved at night (CCB > 0), both in

summer and winter (**Fig. 5**). CCB was significantly lower in winter than in summer (FEB: CCB =
$0.2 \pm 0.2$ mmol $CaCO_3$ $m^{-2}$ $h^{-1}$, SEP: $1.2 \pm 0.6$ mmol $CaCO_3$ $m^{-2}$ $h^{-1}$; t-test: t = -5.2, df = 11.7, *p* =
0.0002). In winter, adding $CO_2$ decreased CCB by more than 80 % in pools C, D, and E (**Fig. 5**).
The $CO_2$ addition even resulted in a transition from a positive community calcification balance to
dissolution in pool C (133 % change, from +0.5 to -0.2 mmol $CaCO_3$ $m^{-2}$ $h^{-1}$). For the two other
pools (A and B), winter $CO_2$ addition increased their relatively small calcification balance (A: +87
%, from 0.1 to 0.2 mmol $CaCO_3$ $m^{-2}$ $h^{-1}$ and B: +71 %, from 0.2 to 0.3 mmol $CaCO_3$ $m^{-2}$ $h^{-1}$). In
summer, changes in CCB due to treatment appeared minimal in pools A, B and E (< 15 % change)
and either increased (C: +67%) or decreased (D: -57%) in the two other pools. In analyses not
presented here, when this budget takes into account winter night:day duration (14:10), all the
pools' budgets switch to net dissolution in future conditions.

## 5/ The particular case of September 2020 tides

During diurnal tides of September 2020 (high PAR and high temperature summer conditions), we
observed an unexpected phenomenon: dissolution occurred at extremely high $pH_T$ values (9-10)
in pools C and E (**Fig. 6**). Under these conditions effectively all the seawater DIC in these pools
was consumed by photosynthesis and calcification (DIC ≈ 0 mmol $kg^{-1}$) four hours after emersion.
As such, the $CO_3^{2-}$ concentration was also effectively zero and the pools reached very low
saturations states ($\Omega_a \approx 0$) despite high pH (**Fig. 6**). These conditions were quickly followed by
indicators of $CaCO_3$ dissolution (increasing TA and DIC) instead of the expected diurnal
precipitation. It is therefore noteworthy that dissolution may happen at high pH, and that pH and



Ω can decorrelate (**Fig. 7**) in situations with high photosynthesis and limited mixing of water
masses.

## DISCUSSION

Temperate tidal pools are environments of extreme variability. In our pools, we observed seawater
temperatures that could increase by up to 10°C in a few hours compared to the adjacent ocean.
During diurnal emersion periods, oxygen concentrations doubled and pH could increase to pH 10
in present-day summer conditions. At night, pH routinely reached levels usually used as the
"treatment" for ocean acidification perturbation experiments (~7.6). Organisms present in the tidal
pools may therefore already be adapted or acclimatized to extreme variability in pH and saturation
state, which could affect their responses to ocean acidification (Andersson et al., 2015). For
example, CCA from a site with naturally high $p$CO$_2$ variability calcified ~50 % more than
individuals from a nearby site of low variability when submitted to oscillating high $p$CO$_2$
treatments (Johnson et al., 2014). Here we show that, even in intertidal communities likely already
acclimated or adapted to variable conditions, with potentially large phenotypic plasticity,
acidification can still modify net community production and calcification rates.

### Diurnal fertilization under CO$_2$ addition

Adding CO$_2$ to simulate future seawater acidification in the pools led to a diurnal fertilization
effect. This increase in the community's net primary production by 20% was particularly visible
in summer (+ 35%), at higher temperatures/metabolic rates. Adding CO$_2$, we also added substrate
for photosynthesis in the form of DIC (**Fig. 3B**) that the algae of the pools can assimilate,
potentially supporting higher DIC use and algal primary production. This effect was apparent from
the start of the emersion, suggesting a direct effect of increasing DIC concentration in the pools.
It seems that photosynthesis in the pools was carbon-limited and that carbon addition therefore
enhanced primary production, in winter and to an even greater extent in summer. During
photosynthesis, the uptake of inorganic carbon leads to a significant decrease in DIC - even in
present-day conditions. Intertidal algae are typically adapted to this with coralline algae in
particular containing CCMs (CO$_2$ concentrating mechanisms) that allow them to achieve primary
production in low DIC concentrations (Raven, 2011). Increasing seawater DIC may however


promote an increase in active and/or passive $CO_2$ and $HCO_3^-$ fluxes towards photosynthetic
compartments. Borowitzka (1981) found that the photosynthetic rate of an intertidal CCA was
highest at pH 6.5 to 7.5 (increased from pH 8.1), a change in pH that was achieved using HCl,
suggesting that increased photosynthetic activity could also be linked to proton gradients/pumps
and/or decreased energy expenditure needed to operate CCMs rather than directly related to $CO_2$
gradients or higher substrate availability.
In winter and summer, pools in present-day and future conditions were autotrophic at
emersion ($NCP_D > CR_N$, **Fig. 5**). If we consider the CPB as integrated diurnal NCP and nocturnal
CR over 24 hours (assuming equal day:night duration), this means that the pools always fixed
more carbon during the day than they respired at night at emersion (NCP >> CR), regardless of
treatment. One methodological uncertainty we highlight regarding net production is that diurnal
DIC-derived NCP estimations were 50 % higher than $O_2$-derived NCP estimates (**Fig. 3C and Fig.**
**4**, $NCP_{DIC} = 1.6 \pm 0.05\ NCP_{O2}$ by day; $R^2 = 75$ %). This discrepancy was far less apparent during
nights, when methods agreed on respiration rates ($CR_{DIC} = 1.0 \pm 0.09\ CR_{O2}$; $R^2 = 56$ %). While
$O_2$-derived NCP appears accurate during the night, $O_2$ production during the day is likely to have
been underestimated due to degassing (e.g., visible formation of oxygen bubbles at the surface of
algae, >150 % air saturation by day vs. < 100 % at night). Thus, estimating diurnal net production
using oxygen measurements may not be appropriate in algae-dominated environments such as
these tidal pools. Nevertheless, despite the difference in absolute NCP estimates, both approaches
indicate a diurnal fertilization effect.

**Nocturnal dissolution under $CO_2$ addition**

In the present study, natural mesocosms - temperate coralline-dominated tidal pools - were used
to investigate the effect of ocean acidification on net calcification at the community level. As we
observed a fertilization effect of $CO_2$ addition by day, we could have expected that it would also
enhance diurnal calcification – as photosynthesis and calcification are tightly linked (Martin et al.,
2013; Martin et al., 2013; Williamson et al., 2017) -, but this was not observed. Treatment had no
significant effect on the daytime net calcification rates, and diurnal variability in calcification
appears to be predominately driven by PAR, temperature, and metabolic activity (NCP). Increasing
metabolic rates - in turn increasing calcification rates - may have however counterbalanced any
calcification suppression or increased dissolution due to acidification, making its effect invisible.



Noisette et al. (2013) similarly found no effect of $p$CO$_2$ treatment on light calcification for *E.*
*elongata*. However, the authors reported a significant decrease in light calcification in *L.*
*incrustans*, net calcification even switched to net dissolution in 750 and 1000 µatm $p$CO$_2$
treatments. While our "future" treatments started at $p$CO$_2$ levels higher than 1000 µatm, the fact
that CO$_2$ addition did not influence diurnal calcification could also be due to favorable saturation
state conditions in the micro-environment in which calcification occurs. The diffusive boundary
layer (DBL) can enhance CaCO$_3$ precipitation micro-environment conditions due to the uptake of
CO$_2$/DIC for photosynthesis. For instance, in light conditions, CCA surface pH has been shown to
reach as high as 8.6 (Houlihan et al., 2020) in surrounding seawater at pH 7.7 (+1.1 pH units),
which would be highly favorable to calcification. But more complex interactions may also be at
work, e.g., CCA may use increases in HCO$_3^-$ (due to CO$_2$ dissolution) to calcify, making them
more resistant to ocean acidification, as suggested by Comeau et al. (2013).
There was net CaCO$_3$ dissolution in the pools at night (-0.7 mmol CaCO$_3$ m$^{-2}$ hr$^{-1}$), even
when waters were still supersaturated with regards to aragonite under present-day conditions.
Night dissolution may be a sign that the DBL of the calcifiers inhabiting the pools is
undersaturated, possibly as a result of respiration. Indeed, Houlihan et al. (2020) observed that
nocturnal algal respiration by CCA, increased CO$_2$ in the DBL, decreasing pH of the DBL by 0.1
units. Such a small pH decrease is however unlikely to explain alone an undersaturation of the
calcifying environment as aragonite saturation state was still above 1.2 in most of the "present-
day" conditions. However, given the solubility of high-Mg calcite - the mineral composing *L.*
*incrustans* and *E. elongata* in particular (Ries, 2011) - can be twice that of aragonite (Sulpis et al.,
2021; Yamamoto et al., 2012), it is possible that undersaturation already occurs at night for this
mineral even for $\Omega_a > 1$. Adding CO$_2$ (from 445 to 1500-2000 µatm) at the start of emersion
significantly increased net dissolution (NCC$_{lm}$), by 40 % in summer and 70 % in winter. In a
previous single-species experiment, Noisette et al. (2013) demonstrated that - for *L. incrustans*
from an area close to our site – dark dissolution doubled with increasing $p$CO$_2$ (1000 µatm vs. 380
µatm), unlike *E. elongata*, for which there was no effect of $p$CO$_2$: the ACA even calcified in the
dark up to 750 µatm (see also similar results from Egilsdottir et al., 2013). Since *L. incrustans* is
the major calcifying species of the tidal pools we studied, it is likely this species drives the results
we observed at the pool community scale. Regardless of the treatment, nocturnal net dissolution





rates (NCC) were also significantly correlated with $\Omega_a$, results similar to those found by
Kwiatkowski et al. (2016) in temperate tidal pools of California, without $CO_2$ addition.

In summer, pools in present-day and future conditions were precipitative (CCB > 0),

meaning that diurnal net calcification exceeded nocturnal net dissolution, regardless of treatment.
Adding $CO_2$ in summer did not consistently change CCB, with most pools showing little change
in CCB due to treatment. By contrast, in the colder winter, the calcification budget was at least 50
% lower than in summer ("present"), with some pools that had comparable net calcification during
the day to net dissolution at night. During this season, adding $CO_2$ had variable impacts on CCB,
decreasing it in three of the pools by more than 80 % and increasing relatively small CCB in two.
These variable effects may be due to differences in community composition and highlight the
difficulty in generalizing the results of natural mesocosm manipulations in which the initial
community composition is not controlled. Nevertheless, we expected $CO_2$ addition to have a
greater negative effect in winter (more dissolution) than in summer, with saturation states being
lower due to colder temperatures, making it more of a "crucial"/ "bottleneck" season. This
emphasizes the need to study the effect of ocean acidification across seasons and temperature
ranges, especially given the associated changes in algal community composition and metabolic
activity.

**Instances of aragonite undersaturation at high pH**

An unexpected phenomenon happened in the pools C and E in summer: although we measured
very high pH values, we observed that total alkalinity suddenly increased, a sign of fast net
dissolution. When we then computed the carbonate chemistry, the saturation states were
surprisingly low ($\Omega_a = 0$ for $pH_T = 10$), which was due to near-zero DIC concentrations – and thus
near-zero $CO_3^{2-}$ concentrations. In these particular conditions, which occurred towards the end of
the tidal emersion period, any $CaCO_3$ precipitation was less than dissolution; precipitation may
even have been impossible due to a lack of DIC substrate. In intertidal pools with a high density
of *Zostera marina*, Miller & Kelley (2021) observed a similar decoupling between pH and $\Omega_a$ with
increases in pH not leading to an increase in saturation state at high pH values due to a lack of
DIC/$CO_3^{2-}$. In our study, we observed even more drastic decoupling between expected changes in
pH, $\Omega_a$ and NCC, with some of the fastest net dissolution rates observed at very high pH and very
low $\Omega_a$ values that were a consequence of near complete consumption of DIC by community



production (**Fig. 7**). Macroalgae cultivation has been proposed as a method of bioremediation to local acidification, in particular to improve aquaculture environments (e.g., Bergstrom et al., 2019; Gao & Beardall, 2022): increase in algal or marine plant cover would reverse or buffer the negative effects of acidification on heterotroph calcifiers. Our results and those of Miller & Kelley (2021) suggest that phytoremediation should not consider pH as the sole indicator for "acidification remediation", and that periodical decreases in saturation state in macroalgae- or seaweed-dominated environments in summer (and during marine heatwaves), may need to be considered for these proposed types of remediations.

**Conclusion**

Relative to its area, human societies are disproportionately reliant on the coastal ocean for the provision of natural resources and climate regulation. Yet our understanding of how anthropogenic carbon emissions and associated ocean acidification will influence natural coastal ecosystems and community metabolism remains limited. In the present study, we manipulated the carbonate chemistry of natural temperate intertidal pools to explore the potential impact of future ocean acidification on community-level calcification and production. We find evidence of large seasonal, diel and community-specific differences in the sensitivity of intertidal community metabolism to acidification. Diurnally, acidification was found to enhance net community production, with this "fertilization effect" indicating algal photosynthesis is naturally carbon limited in such environments at emersion. Diurnal net community calcification was unaffected by acidification. In contrast, nocturnal acidification resulted in greater net community dissolution in the intertidal pools yet had no consistent effect on community respiration. Integrated over day/night emersion periods, the intertidal mesocosms maintained positive net community calcification and production under both present-day and future conditions. Albeit considerable differences between individual pools and strong seasonal dependencies, our results indicate that the net calcification and production of temperate intertidal communities - likely acclimated/adapted to variable conditions - could be affected by future acidification.

**ACKNOWLEDGEMENTS**



We thank Elsa Perruchini, Léonard Dupont, Corentin Clerc, Priscilla Le Mezo, Alban Planchat,
Maud Chevalier, Anne Cornillon, Annabel Antheaume, Maïlys Roux and Clarisse Dufaux for their
kind assistance with fieldwork. This project is fully funded by the CHANEL research chair:
*Understanding the Linkages between the Ocean, the Carbon Cycle, and Marine Ecosystems under*
*Climate Change*. Data presented for adjacent Atlantic waters characteristics (main text and the
supplementary material) were kindly provided by the SOMLIT network database (Service
d'Observation en Milieu Littoral; www.somlit.fr) on June 2022.

**AUTHORS CONTRIBUTIONS**
ND, SM and LK designed the experiments and ND carried them out with help from all co-authors.
ND analysed the data and prepared the manuscript, with contributions from all co-authors.

**COMPETING INTERESTS**
The authors declare that they have no conflict of interest.

**DATA AVAILABILITY**
Raw data and linear regression model results are provided as supplementary in the Appendix.



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



# 1 Figures with legends -

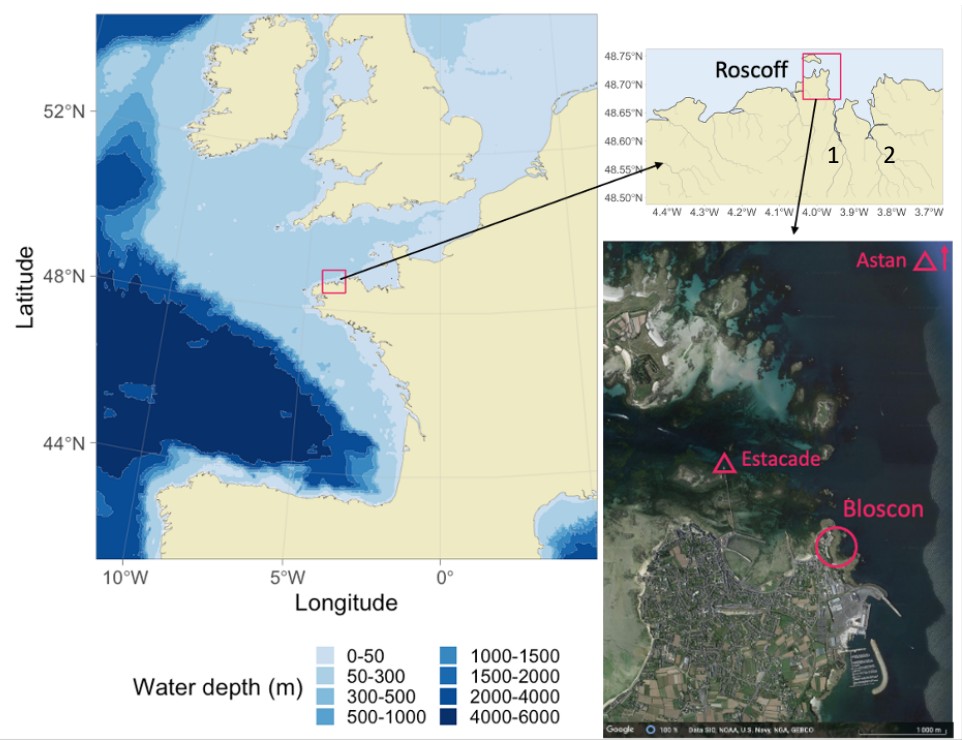

**Figure 1 - Field site location** on a map of Europe (**left**). The study site (Bloscon) is located in Roscoff,
Brittany, France (**right**, top: river mapping data from *HydroSHEDS,* **1**. Penzé river and **2**. Morlaix
river; bottom: satellite image from © *Google Earth*: earth.google.com/web/, acquired in June 2022).
The SOMLIT stations Astan and Estacade are indicated with triangles (www.somlit.fr).

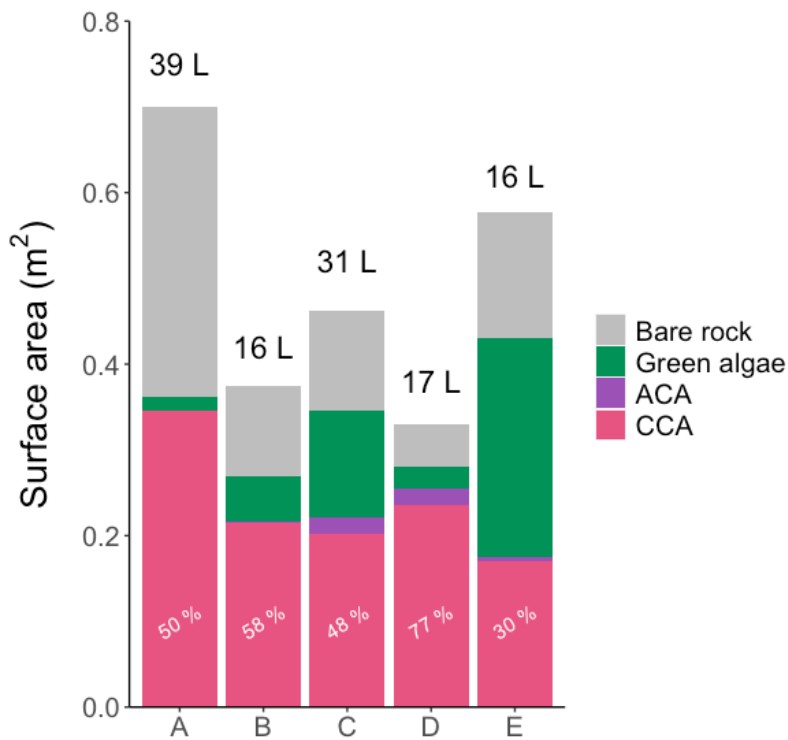

**Figure 2 - Pool area, volume and coverage -** Surface of the five pools (A-E, September 2020) covered
by crustose coralline algae (CCA, pink), articulated coralline algae (ACA, purple) and green algae
(green) or free of algae ("bare rock", grey). The length of the bars represents total pool surface area
($m^2$) and the volume of each pool (L) is indicated above. The relative coverage (%) of calcifying algae
(ACA + CCA) in each pool is given. Details for the other seasons are available in **Supp. Mat. Pools**
**Fig. SP1** and **SP2**.



**Figure 3 - Composite daily pool conditions and biological activity for all pools. A) temperature**
(°C), **salinity and oxygen concentration** (mg L$^{-1}$) **and Photosynthetically Active Radiation** (PAR,
μmol m$^{-2}$ s$^{-1}$), **B) pH$_T$, Total Alkalinity** (TA, μmol kg$^{-1}$), **$p$CO$_2$** (μatm), **dissolved inorganic carbon**
(DIC, μmol kg$^{-1}$) **and aragonite saturation state** ($\Omega_a$), and **C) DIC and O$_2$-derived NCP or CR**
(mmol C or O m$^{-2}$ hr$^{-1}$) **and NCC** (mmol CaCO$_3$ m$^{-2}$ hr$^{-1}$).  Colors represent seasons (**A:** blue for
February, orange for April, red for September) and treatment (**B and C**: purple for "present" and green
for "future"). Horizontal dotted grey lines represent the mean values of the adjacent ocean. Curves
were fitted by season for PAR and for diurnal NCP and NCC using a local polynomial regression
(*loess*) with 95% confidence interval. Number of observations: n = 1551 for temperature, salinity and
pH$_T$, n = 1169 for oxygen concentration (data recorded < 22 mg L$^{-1}$) and n = 632 (hourly data) for the
carbonate chemistry parameters, NCC and NCP or CR (**B**). All pools are shown.

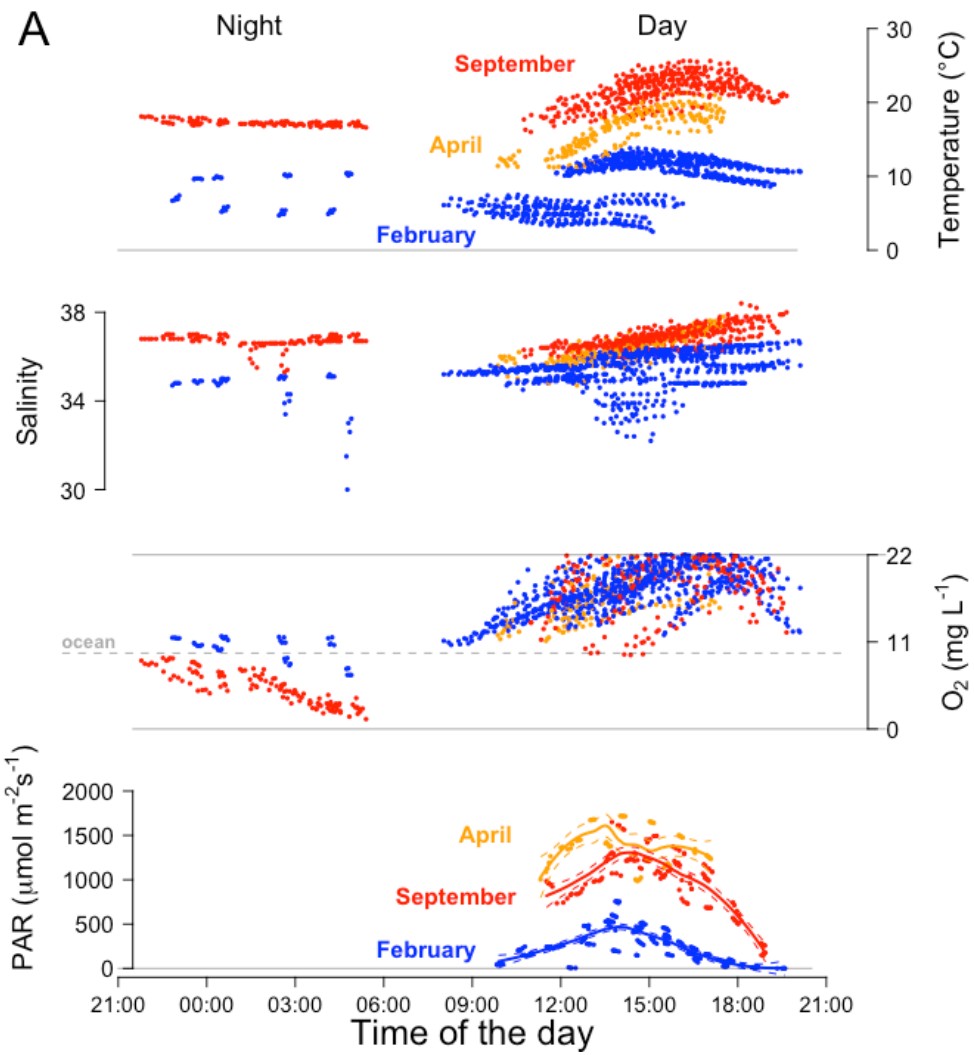



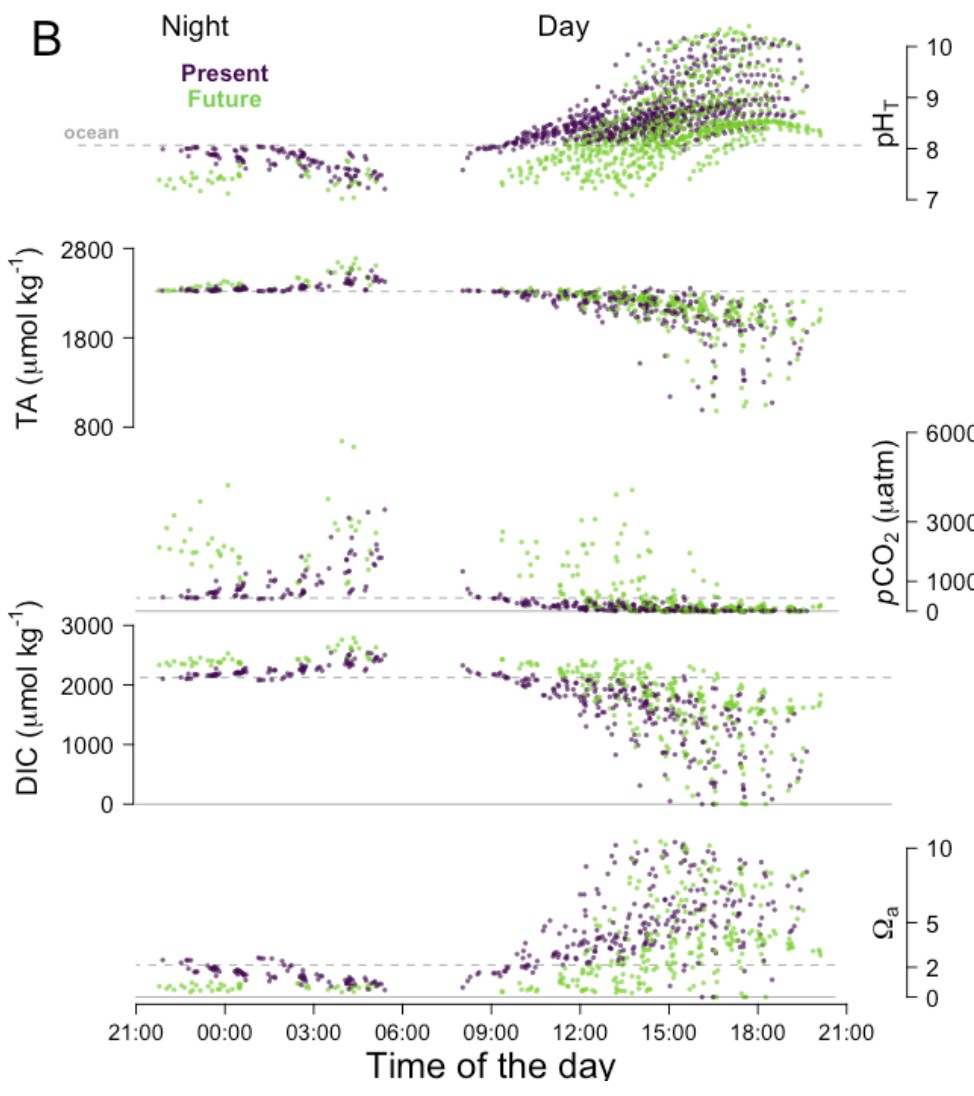



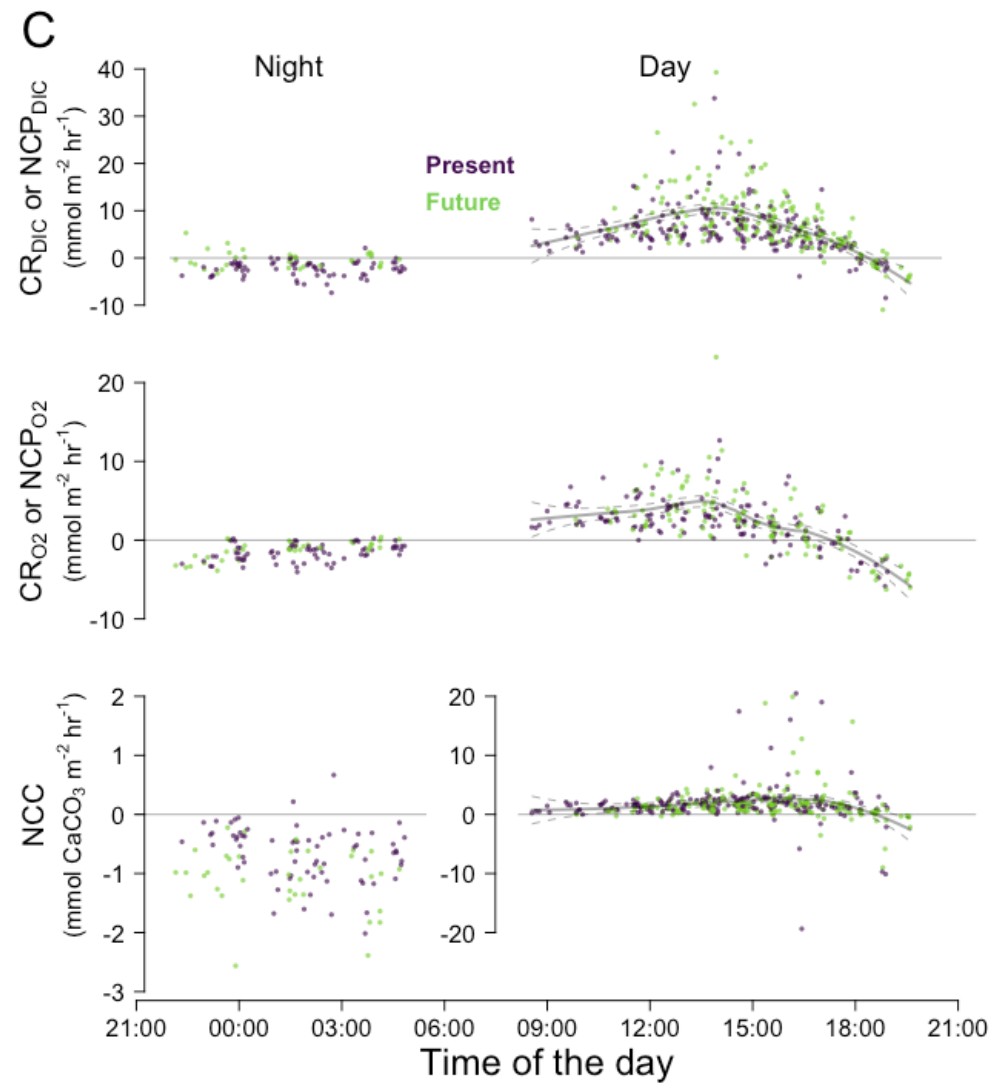





**Figure 4 – A) O$_2$-derived (white boxes) and DIC-derived (colored boxes) NCP$_{lm}$ (mmol m$^{-2}$ hr$^{-1}$),**
**and B) NCC$_{lm}$ (mmol CaCO$_3$ m$^{-2}$ hr$^{-1}$) during the day and night (shaded areas), by season and**
**by treatment** (purple for "present" and green for "future") – Rates are presented as boxplots showing
median, 1$^{st}$ and 3$^{rd}$ quartile and 1.5 inter-quartile range (bars), with overlayed individual observations
(round symbols). Individual rates were calculated for each pool, each tide and each treatment: n = 50
(FEB-day), n = 10 (FEB-night), n = 10 (APR-Day), n = 25 (SEP-Day), n = 20 (SEP-Night). Seasons:
FEB for winter (pooled February 2020 and 2021), APR for spring (April 2021) and SEP for summer
(pooled September 2020 and 2021). *Note that for NCC$_{lm}$, nights (<0) and days (>0) have different y-*
*axis scales for better visualization of night differences. Statistical details of the linear regressions can*
*be found in the corresponding **Supplementary Materials**. For O$_2$-derived NCP$_{lm}$, in September, three*
*rates were out of the range plotted and their values are indicated next to the small arrow.*

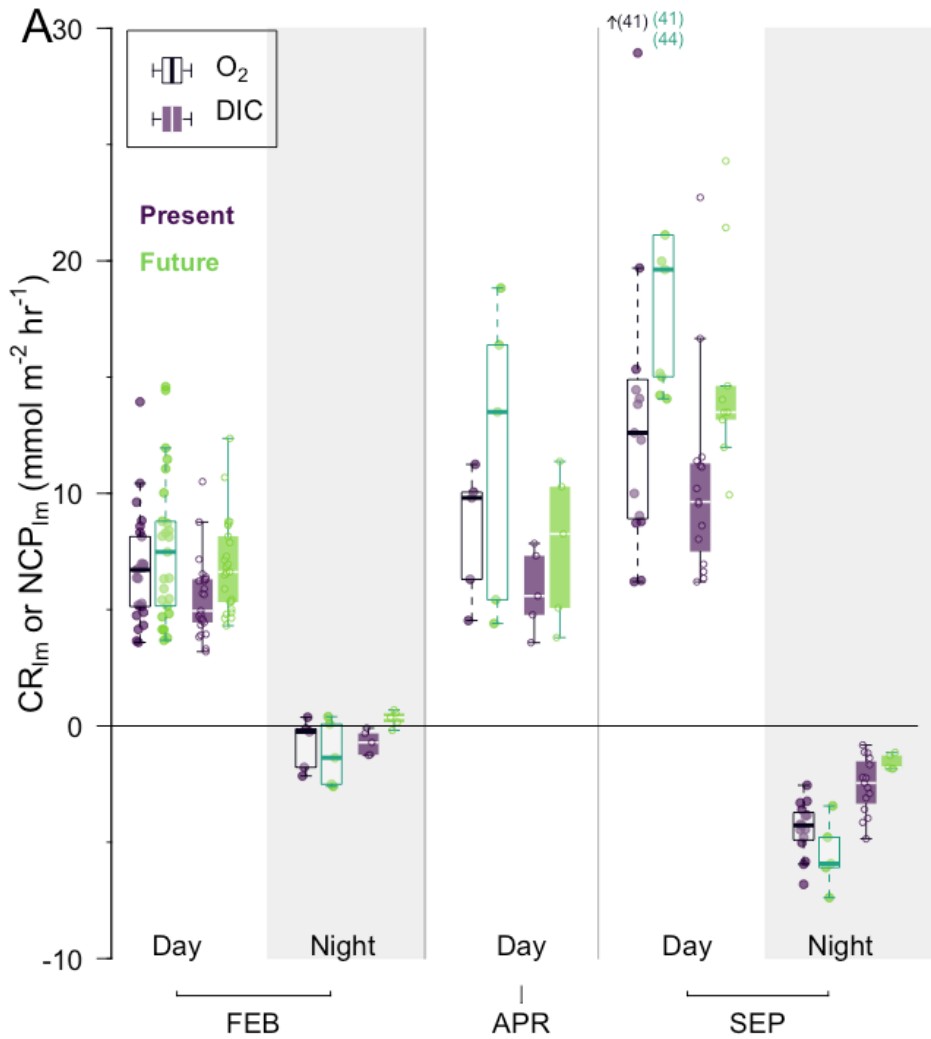



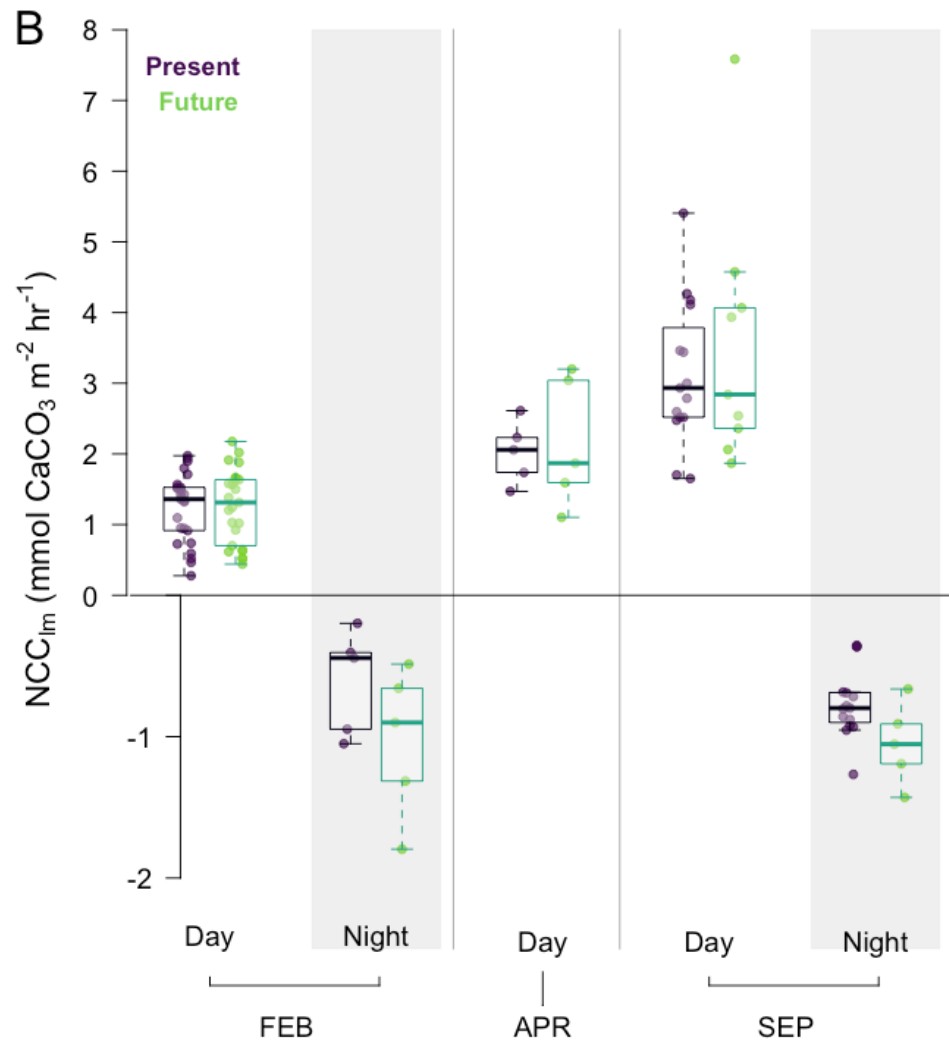



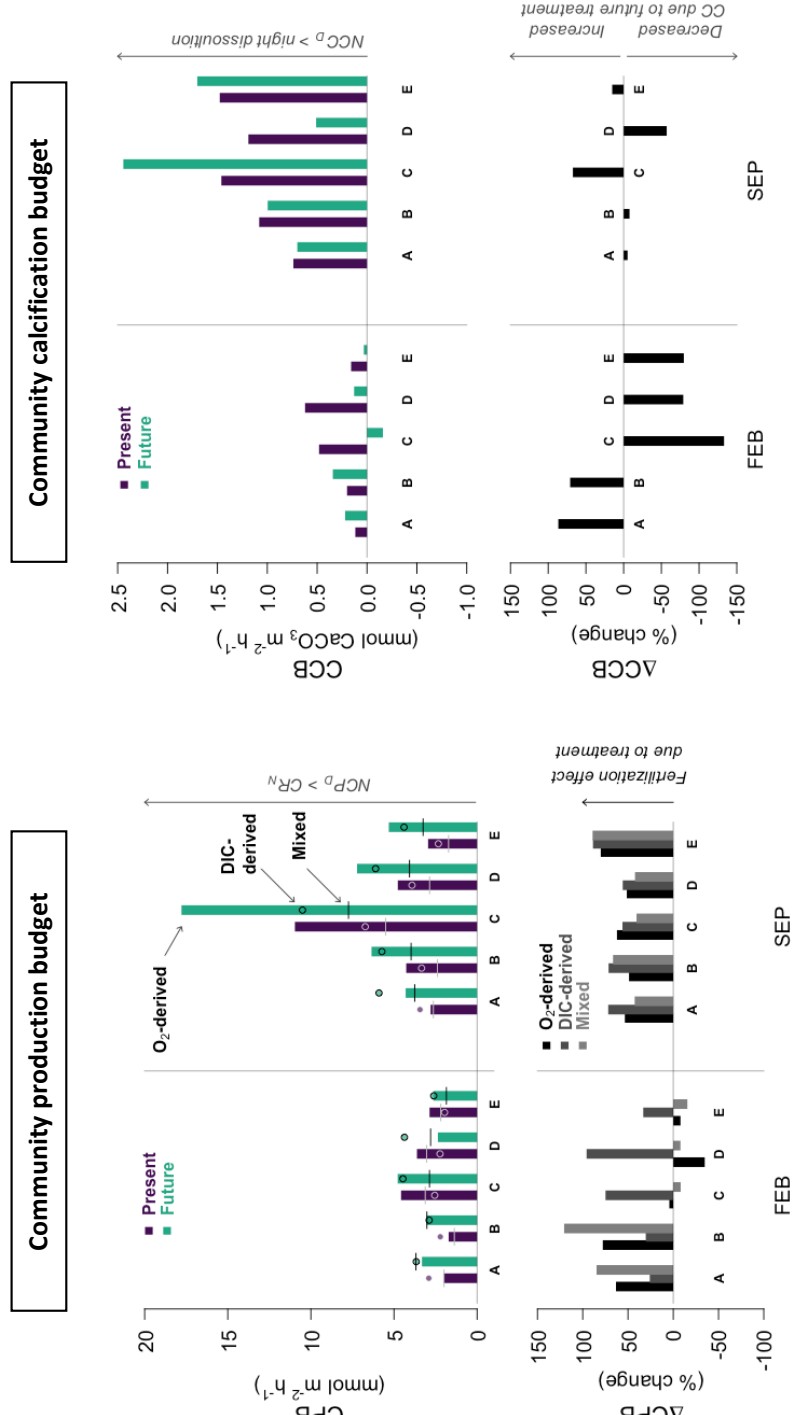

**Figure 5 – Community production budget: CPB [1], and calcification budget: CCB (upper right panel, mmol CaCO$_3$ m$^{-2}$ hr-1)** by treatment (purple for "present" and green for "future") for each pool and season (same legend as Fig. 4). CPB >0 if diurnal NCP > nocturnal respiration and CCB > 0 if diurnal NCC > nocturnal dissolution. CPB was estimated three different ways: from O$_2$-derived NCP (bars), from DIC-derived NCP (round symbols) and from nocturnal O$_2$-derived CR combined with diurnal DIC-derived NCP ("mixed", vertical segments). **The bottom panels present the change (%) of diel production (ΔCPB: left) and diel calcification (ΔCCB: right) due to CO$_2$ addition.** Positive ΔCPB indicates a fertilization effect due to the CO$_2$ addition; negative ΔCCB is expected if the CO$_2$ addition decreases net calcification/increases net dissolution. All three methods to estimate CPB indicate a fertilization effect in summer.

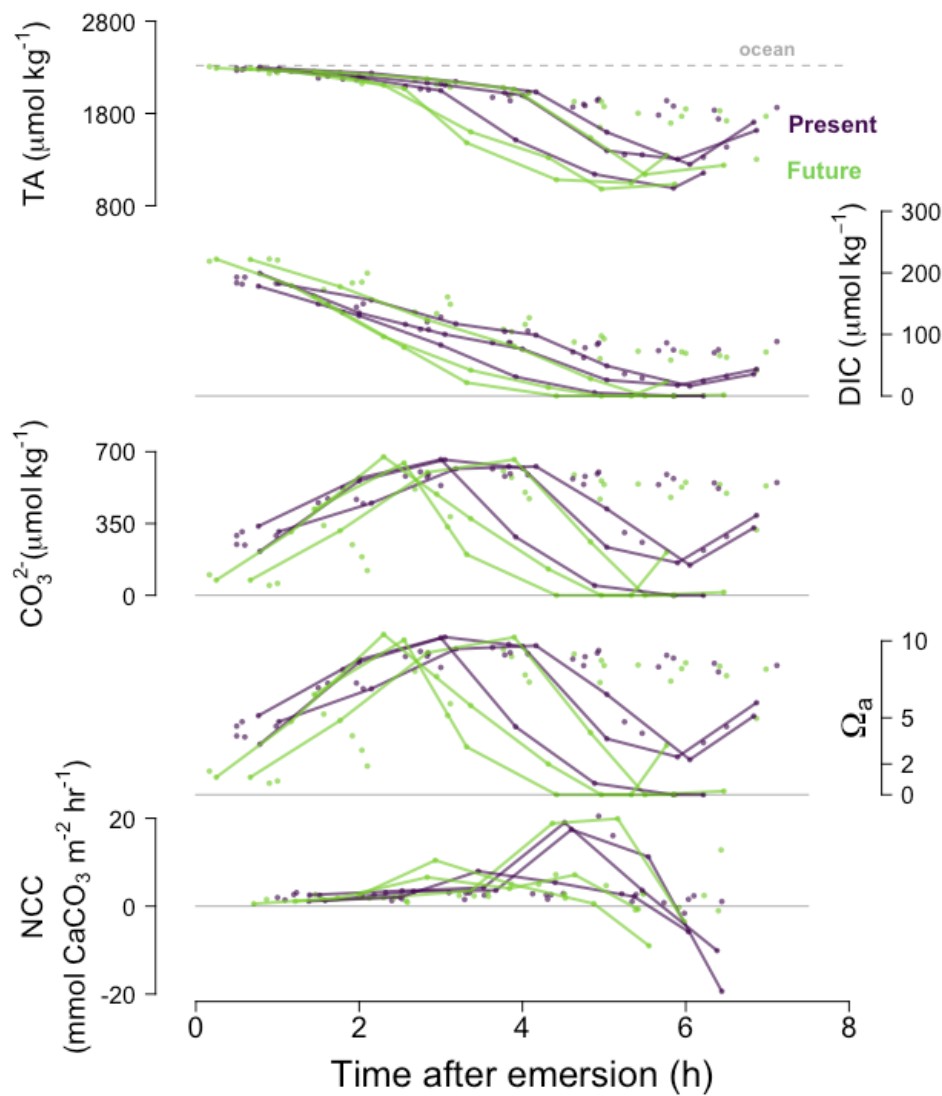

50

**Figure 6 - Time series for September 2020 diurnal data only:** **A) Total Alkalinity (TA, μmol kg$^{-1}$), dissolved inorganic carbon, $CO_3^{2-}$ concentration (μmol kg$^{-1}$), aragonite saturation state ($\Omega_a$) and NCC (mmol m$^{-2}$ hr$^{-1}$) with time after emersion, by treatment** (purple for "present" and green for "future"). The lines in bold represent individual pools C and E that switched from calcification to dissolution when pH$_T$ was still above 9. A similar figure in **Supp. Mat. (Fig. S4)** shows that sunset/irradiance are not correlated with the sudden change towards dissolution.

57

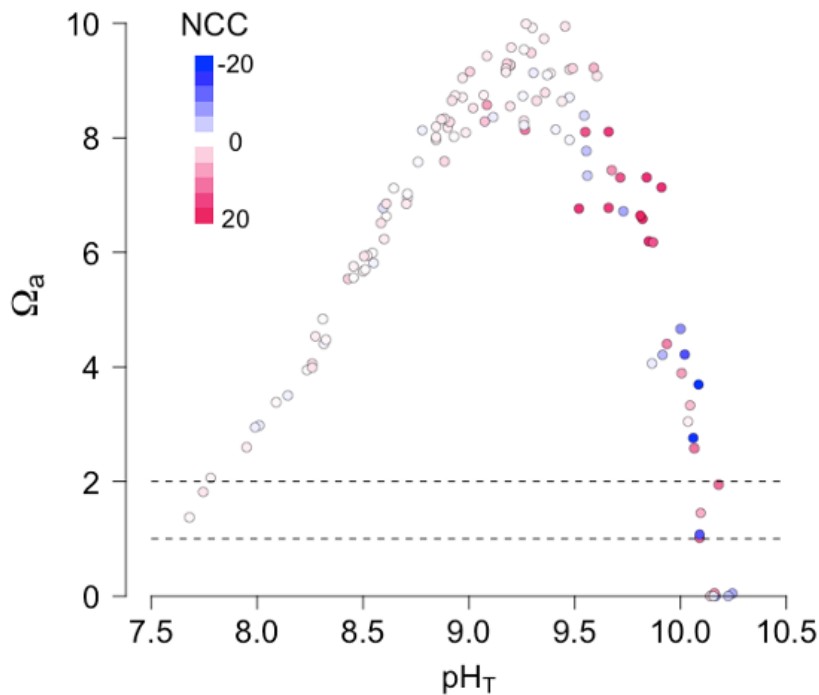

**Figure 7 – At very high pH there was both fast net calcification (red) and rapid net dissolution (blue)**: In some extreme cases, $pH_T$ was not a good indicator or seawater saturation state ($\Omega_a$). Selected dataset of diurnal low-tide emersion periods from September 2020. Colors represent NCC (in mmol $CaCO_3$ $m^{-2}$ $hr^{-1}$, as presented in **Fig. 3C**). Dashed horizontal lines represent saturation state for aragonite ($\Omega_a = 1$) and for high-Mg calcite ($\Omega_a = 2$).