# Peer review of "Ocean acidification enhances primary productivity and"

_Biogeosciences, 2023_

## Author Comment (AC1)

**Answers to reviewers, Dorey et al., BGD (bg-2023-79).**

We would like to thank the reviewers for their constructive criticism and the time they spent on careful proofing of our submitted manuscript. Answer to the reviewers can be found below in blue: The lines mentioned refer to the original version.

**RC1**: 'Comment on bg-2023-79', Zvi Steiner, 29 Jun 2023 reply

*Acidification of the ocean due to the ongoing anthropogenic $CO_2$ emissions is affecting calcification and photosynthesis in the ocean. However, the effect of acidification on the community level is complex as different organisms might respond differently to this issue. Dorey et al. studied the effect of ocean acidification in natural rock pools, which is an interesting approach as the rock pools are isolated from the ocean during low tide forming a perfect mesocosm experiment with a natural benthic community. Conditions in the rock pools naturally vary greatly between day and night and between seasons, hence the community is expected to be fairly resilient to acidification, yet the experiments show a clear effect of the interventions.*

*My main concern about the paper is that it lacks a proper inorganic control. The assumption of the authors is that there is no equilibration of $CO_2$ added to the pools with atmospheric $CO_2$. This is a fair assumption in the open ocean, as it takes $CO_2$ several years to equilibrate with the atmospheric pressure but I'm not sure it holds in the experimental setup because of the large deviation of the $pCO_2$ in the treated pools from atmospheric values, fluctuations in pool temperature during the day (particularly in summer), and the large surface area and limited volume of the pools that are also exposed to the elements. Therefore, I think it is important to add a control experiment that will test if there is a change in seawater $pCO_2$ after a $CO_2$ enrichment at the study site in summer conditions. The control experiment can be done in a simple tub with similar dimensions to one of the pools, that is initially filled with artificial or filtered seawater and treated similar to the $CO_2$ enrichment experiment. It might also be useful to bubble $N_2$ into a second identical tub to stimulate low $CO_2$ and $O_2$ conditions and test the rate of penetration of these gases during a few hours.*

Authors: We thank the reviewer for pointing this study limitation out and while we are unable to do the proposed additional experiment for the moment, we will present below why we consider the $CO_2$ air-sea diffusive transport to be negligible using the theoretical fluxes of $CO_2$ in a 0.5 $m^2$ pool treated with additional $CO_2$ (+1500 µatm).

First, we need to explain why we do not consider the effects of wind and waves in the air-sea gas exchanges in our tidal pool setting.
The effect of wind is very minimal on the pools, as the pools are well-protected from the wind by higher rocks on the floor, in a boundary layer where surface friction is high. Furthermore, with its eastern orientation, the Beach of Bloscon is generally sheltered from the usual western and south-western winds (see for example the figure below presenting the wind direction and figure 2.3 in nearby Brest in the thesis here:

https://www.researchgate.net/publication/286473561_Dynamique_et_echanges_sed imentaires_en_rade_de_Brest_impactes_par_l%27invasion_de_crepidules/figures?lo =1, or in a review by the national meteorological institute *Météo France*, here: https://donneespubliques.meteofrance.fr/donnees_libres/bulletins/BCMR/BCMR_05_ 201401.pdf).

[Figure]

There were no movement/waves in the pools, nor was there any visible breeze waves. The effect of surface wave on gas transfer velocity is generally neglected in small lakes and even though, it is discussed for larger lakes such as Lake Geneva (e.g., Perolo et al. in ESD, 12, 1169–1189, 202, https://esd.copernicus.org/articles/12/1169/2021/), it can reasonably be neglected in our small tidal pools.

There is no vertical or horizontal transport of $CO_2$ due to convection in those small pools since we are only interested in them when they are closed at low tide.

We can thus consider the flux of $CO_2$ at the surface of the tidal pools is only due to net diffusive transport:

$$F_{CO2} = k \; \alpha \; \Delta pCO_2$$

Where $F$ is the $CO_2$ flux ($\mu mol \; cm^{-2} \; h^{-1}$), $\alpha$ is the $CO_2$ solubility coefficient ($\mu mol \; cm^{-3} \; atm^{-1}$), $k$ is the gas transfer velocity ($cm \; h^{-1}$), and $\Delta pCO_2$ is the gradient of partial $CO_2$ pressure ($pCO_2$) between the water and the air (see example references for this equation in Perolo et al., 2021, in ESD, 12, 1169–1189, 202, https://esd.copernicus.org/articles/12/1169/2021/ ). We also consider that there is no need to correct for atmosphere (pressure = 1atm) as we are at sea level at Paris latitude.

- For the solubility coefficient of $CO_2$ , a table depending on water temperature and salinity can be found in Weiss (1974) Mar Chem, 2 (3), 203-215

([https://www.sciencedirect.com/science/article/pii/0304420374900152](https://www.sciencedirect.com/science/article/pii/0304420374900152)): at a temperature 20°C and a salinity of 35, **α = 3.32 $10^{-2}$ mol $L^{-1}$ $atm^{-1}$.**

- For the ***ΔpCO₂*** we take the average treatment of +1500 µatm compared to a "control" pool (i.e., a pool equilibrated to the atmosphere where Δ $pCO_2$ = 1). ***ΔpCO₂* = 1.5 $10^{-6}$ atm.**

- For the gas transfer velocity, we used the data and personal communication from David Ho (Ho et al., 2018 *On factors influencing air-water gas exchange in emergent wetlands.* Journal of Geophysical Research: B 123, 178–192, [https://agupubs.onlinelibrary.wiley.com/doi/epdf/10.1002/2017JG00429](https://agupubs.onlinelibrary.wiley.com/doi/epdf/10.1002/2017JG00429)). David Ho that told us that, he had found an estimation of ***k* = 1 cm $h^{-1}$** the method in small artificial pools. The result $k$ = 1 cm $h^{-1}$ is also what is presented in baseline field conditions in the Everglades the paper Ho et al. (2018) cited above.

Therefore:

$$F_{CO2} = k \; \alpha \; \Delta pCO_2$$

$$F_{CO2} = 1 \text{ cm } h^{-1} \times 3.3 \; 10^{-2} \text{ mol } L^{-1} atm^{-1} \times 1.5 \; 10^{-6} \text{ atm}$$

$$F_{CO2} = 1 \text{ cm } h^{-1} \times 3.3 \; 10^{-5} \text{ mol } cm^{-3} atm^{-1} \times 1.5 \; 10^{-6} \text{ atm}$$

$$F_{CO2} = 4.95 \; 10^{-11} \text{ mol } cm^{-2} h^{-1}$$

$$F_{CO2} = 0.495 \; \mu\text{mol } m^{-2} h^{-1}$$

In our study, typical pool size was 0.5 $m^2$, or 5000 $cm^2$. Thus, the seawater of a treatment pool (+1500 µatm) "loose" an approximative **4.95 $10^{-11}$ mol $cm^{-2}$ $h^{-1}$ x 5000 $cm^2$** = 2.5 $10^{-7}$ mol $h^{-1}$ to the atmosphere, or **0.25 µmol per hour due to net diffusive transport**.

In the present study (see Figures 3), the biological rates outweigh by a factor 100 this net diffusive transport. For instance, calcification rates (a rate calculated using total alkalinity, not $DIC/CO_2$ concentrations) range from -2 to +4 mmol $m^{-2}$ $h^{-1}$, i.e., a change by -1 to +2 mmol per hour in a 0.5 $m^2$ pool. Compared to calcification, the change in carbon due air-sea exchanges is thus less than 0.1%.
To put this result - a change by 0.25 µmol per hour due to net diffusive transport - further in perspective, data we obtained using individual incubations conducted in the same pools conditions in summer, that gastropods (*Gibbula pennanti*) respire at rates of 10 µmol $h^{-1}$ per individual.

Even when considering higher values for $k$ and α, it seems unlikely that the air-sea flux would play a major role in the observed changes. For instance, in winter, water at 5°C would only multiply the gas solubility by 2 (**α = 5.4 $10^{-2}$ mol $L^{-1}$ $atm^{-1}$**). In in natural field conditions of the Everglades (Ho et al. 2018), the maximum values found were k = 3 cm $h^{-1}$. In Geneva's Lake, stronger winds (> 5 m $s^{-1}$) led to k values around

30-40 cm h$^{-1}$ (Perolo et al., 2021). However, even these gas transfer velocity levels which would not increase gas exchanges to a considerable level:

$$F_{CO_2} = 40 \text{ cm h}^{-1} \times 5 \text{ } 10^{-2} \text{ mol L}^{-1} \text{ atm}^{-1} \times 1.5 \text{ } 10^{-6} \text{ atm}$$
$$= 2.25 \text{ } 10^{-9} \text{ mol cm}^{-2} \text{ h}^{-1}$$

We thus conclude that the air-sea exchanges were negligible, and that they did not affect the treatments. We also added a sentence in the material and methods to reflect this "The air-sea fluxes due to net diffusive transport are considered to be negligible (see *answer to reviewers*)."

Out of curiosity, and depending on our abilities, we may however conduct an experiment with artificial seawater in early September in Roscoff.

**RC1**: *The discussion about nocturnal dissolution should include more references about supersaturated CaCO3 dissolution in seawater – you can start with Milliman et al., DSR, 1999 and Subhas et al., GBC, 2022, and references therein.*

Authors: We thank the reviewer for pointing out to us those two articles:

Milliman, J. D., Troy, P. J., Balch, W. M., Adams, A. K., Li, Y.H., and Mackenzie, F. T.: Biologically mediated dissolution of calcium carbonate above the chemical lysocline? Deep-Sea Research I, 46, 1653–1669, 1999. https://www.sciencedirect.com/science/article/abs/pii/S0967063799000345

Subhas, Zvi Steiner et al. 2022. Shallow Calcium Carbonate Cycling in the North Pacific Ocean https://agupubs.onlinelibrary.wiley.com/doi/abs/10.1029/2022GB007388

However, after careful reading, we did not know where they would fit in our discussion as they refer to plankton and the open ocean.

*Other comments:*

*Line 283: Data is plural.*

We changed "data was limited to the first three hours of emersion as high O2 concentrations" to "data were limited to the first three hours of emersion as high O2 concentrations".

*Line 331: Is the total alkalinity concentration salinity normalised?*

The total alkalinity presented here was not salinity normalized.

*Line 334: day:night period should be close to 12:12 in September.*

The times are given as a broad indication. Precisely, the sunrise to sunset period was:

- 14:25 to 14:29 hours on 28-29 April 2021
- 10:07 to 10:17 hours on 14-17 February 2020

- 9:50 to 10:30 hours on 9-19 February 2021
- 13:20 to 12:45 hours on 2-11 September 2020
- 13:07 to 12:56 hours on 6-9 September 2021

The time before sunrise and sunset in the summer period we investigated is thus closer to 13 hours. With twilight times being particularly long in September (around 1 hour of light before sunrise/after sunset), we considered that 14:10 for September was a good approximation of "day:night", when considering "night" as "no light".

*Lines 404-5: Not clear, do you refer to the absolute productivity or to the change in productivity due to the interventions?*

Agreed, this sentence was unclear. We transformed "Compared to pool A, productivity was significantly lower in pools B and E and significantly higher in pools C and D ($p < 0.003$)." by "$NCP_{lm}$ was significantly lower in pools B and E than in pool A, and significantly higher in pools C and D ($p < 0.003$)."

*Table 2: It is not clear what do the numbers under "Treat" stand for. Is it the difference between the slopes of the linear regressions of all values from each treatment?*

As explained in the table legend, "The models include three fixed factors: *Temp* (mean temperature: a continuous factor), *Treat* (for $CO_2$ "future" treatment vs. "present", two levels) and pools (vs. A, five levels), and one random effect (low-tide emersion period or the calendar day at which the pool was measured)." And this was also defined lines 312-317.

If the referee refers to the columns' headers, *Estimate* refers to the estimate of each fixed or random variable of the presented model (treatment, temperature or pools), followed by the standard error and *p*-value associated to that estimate. This is standard reporting in glmm, however, if there would be a way to make it clearer, we would happily change it.

*Lines 444-7: Seasonality in DIC is discussed in the text but not shown in Figure 3.*

Yes, this is correct. As stated in the sentences, the detailed data is available in Table S1. However, to make this part more obvious without adding too many figures in the text, we propose to add a figure in the Supplementary Material, similar to Figure 3A that would show the seasonality of pHT TA, DIC, $pCO_2$ and Ω. The figure will be this one presented below:

Figure S3 - Composite daily pool conditions and biological activity for all pools. pHT , Total Alkalinity (TA, µmol kg -1 ), pCO2 (µatm), dissolved inorganic carbon (DIC, µmol kg -1 ) and aragonite saturation state (Ωa). Colors represent seasons (A: blue for February, orange for April, red for September). Horizontal dotted grey lines represent the mean values of the adjacent ocean.

[Figure]

*Lines 505-7: This is a key claim but it contradicts Figure 5. You should show how you came to it or remove this sentence.*

This was also pointed out by RC2 and we removed the sentence entirely.

*Lines 585-9: The effect of pH on calcification is indirect through its effect on the carbonate ion concentration, and in any case is important at the site of calcification but not anywhere around the organism. Do the CCA calcify internally within the cells or externally? If internally, the calcification sites are not necessarily exposed to the ambient pH.*

We thank the reviewer for this remark. We agree that calcification sites are not directly exposed to the ambient pH since coralline algae precipitate high-magnesium calcite within their cell walls, along polysaccharide microfibrils. We have now added the following text to the discussion section:

*"Although there are conflicting results indicating that saturation state of the ambient seawater is a key driver of coralline algal calcification, the biomineralization process in coralline algae has been shown to present a certain degree of biological control (Tomazetto de Carvalho et al. 2017; Nash et al. 2019). Recent work using boron isotopes (δ11B) as a proxy for pH showed that coralline algae have ability to elevate pH at their site of calcification (Cornwall et al. 2017)."*

References:

Cornwall CE, Comeau S, McCulloch M. Coralline algae elevate pH at the site of calcification under ocean acidification. Glob Change Biol. 2017; 00:1–12.

Nash MC, Diaz-Pulido G, Harvey AS, Adey W (2019) Coralline algal calcification: A morphological and process-based understanding. PLOS ONE 14(9): e0221396. https://doi.org/10.1371/journal.pone.0221396

Tomazetto de Carvalho et al. (2017) Biomineralization of calcium carbonate in the cell wall of Lithothamnion crispatum (Hapalidiales, Rhodophyta): correlation between the organic matrix and the mineral phase. https://doi.org/10.1111/jpy.12526

*Lines 590-610: It is also likely that heterotrophic organisms graze some CaCO3 when eating the algae and dissolution occurs within their guts (hence independent of ambient saturation levels). Microbial respiration of organic matter attached to CaCO3 will have the same effect with dependence on saturation states.*

We thank the reviewer for his valuable suggestion. This is well known in pelagic system, where a significant carbonate fraction is effectively lost to dissolution during zooplankton grazing on calcifying phytoplankton. For, example about half of the grazed CaCO3 shells is assumed to dissolve in the guts of grazers (Gehlen et al. 2007; Biogeosciences, 4, 505–519). This could also occur in benthic systems, and in particular in rockpools, where grazing activity is mainly due to patellid limpets (dominant grazer on rocky shores throughout the NE Atlantic). We have now added the following text to the discussion section:

*"As dominant grazer in rockpools, patellid limpets could be particularly active at night (Lorenzen 2007). Their diet is mainly driven by the local availability of food sources (Schaal & Grall 2015) and encrusting coralline algae can be an important food source for these herbivores (Maneveldt et al. 2006). Accordingly, limpets, which are likely to actively consume encrusting coralline algae at night, can have a large percentage of grazed coralline algal CaCO3 in their gut (Maneveldt et al. 2006), which could dissolve easily at night."*

References:

Lorenzen S. (2007) The limpet Patella vulgata L. at night in air: effective feeding on Ascophyllum nodosum monocultures and stranded seaweeds, Journal of Molluscan Studies, Volume 73, Issue 3, August 2007, Pages 267–274, https://doi.org/10.1093/mollus/eym022

Maneveldt et al. (2006) The role of encrusting coralline algae in the diets of selected intertidal herbivores. Journal of Applied Phycology (2006) 18: 619–627. DOI: 10.1007s/10811-006-9059-1

Schaal, G. & J. Grall (2015) Microscale aspects in the diet of the limpet Patella vulgata L. Journal of the Marine Biological Association of the UK 95, 1155-1162. DOI:10.1017/S0025315415000429

*Lines 649-652: These sentences don't belong here, they were not discussed in the paper.*

The sentences the reviewer refers to: "*Relative to its area, human societies are disproportionately reliant on the coastal ocean for the provision of natural resources and climate regulation. Yet our understanding of how anthropogenic carbon emissions and associated ocean acidification will influence natural coastal ecosystems and community metabolism remains limited.*" are the opening on this paper and allude to future research possibilities. This section is meant to connect the significance of the results to a bigger picture. However, if this is not accepted by the reviewer, we can remove these sentences.

**RC2**: 'Comment on bg-2023-79', Erwann Legrand, 30 Jun 2023 reply

**General comments:**

*The manuscript "Ocean acidification enhances primary productivity and nocturnal carbonate dissolution in intertidal rock pools" describes a novel and interesting experimental approach, manipulating the carbonate chemistry of intertidal pools to investigate the effects of ocean acidification on temperate intertidal communities. Overall, this is a well written and valuable paper that significantly contributes to the state of knowledge on the impacts of ocean acidification on marine ecosystems. The introduction addresses the current knowledge, the objectives and the hypothesis of the study. The methods are precise, and the results are well-presented. The discussion highlights the main findings and significance of the study. I can see no major issues with the manuscript, but have a number of suggestions which hopefully can improve the clarity of the paper.*

*My main comment concerns the general conclusion on the vulnerability of tide pool communities to ocean acidification. The experiment was carried out during emersion cycles only, and **it seems important to me to discuss the potential response of tide pool communities during immersion cycles, as environmental conditions differ considerably.** The tide pools studied here were submerged for nearly 12 hours a day. In future conditions of ocean acidification, communities **would therefore be exposed to low pH for longer periods of time. I think this point is important to discuss**.*

**Specific comments:**

*L.50: Please mention the IPCC scenario RCP8.5*

We changed the sentence "Ocean acidification is projected to further decrease average surface pH by up to 0.4 units by 2100 (Kwiatkowski et al., 2020), and is identified as a major threat to marine ecosystems (IPCC, 2019)." to "Ocean acidification is projected to further decrease average surface pH by up to 0.4 units by 2100 (scenario RCP8.5, Kwiatkowski et al., 2020), and is identified as a major threat to marine ecosystems (IPCC, 2019).

*L.164: "the relative area covered by each type of algae were estimated from aerial photographs". Considering the important growth of green algae in some tide pools (especially in September) and the canopy it generates, how did you consider the surface of underlying coralline algae based on aerial pictures?*

This is a valid concern. We added a sentence regarding this problem (line 174): "One limit of this method of aerial photography is that it only takes into account what is visible from above (2D), and these estimates may be biased against algae that were hidden under the green algae canopy in summer or that were in crevices/under rocks".

*L.167: I am a bit confused by figure 2. You state that the surface area ranged from 0.3 to 0.6 m², however it appears to be up to 0.7 m². Also, can you explain why the surface sometimes varies up to ± 0.2 m² among seasons with a similar volume? I suggest keeping figure 2 as supplementary material and only give the main characteristics of tide pools in the text.*

This is true, the text was corrected to $0.7m^2$. The difference in season is likely to be either the precision of the measurement (± 0.2 m²) or due to evaporation (the photos were not always taken at the start of the emersion), we added a note about this in the supplementary material. We think the figure 2 still fits nicely in the main manuscript as it is our only full detailed description of the pools.

*L.168: "CCA: 30 to 71 % of the benthic cover". These numbers differ from fig.2. Did you consider the average during the whole experiment? Please clarify it.*

This is true, the text was corrected to "CCA: 30 to 77 % of the benthic cover".

*L.186: You state that you decreased the pH to 7.5 (same L.137), while L.197, the desired pH level was 7.6. Is it a difference between targeted pH and measured pH?*

Thank you for pointing this mistake out, we replaced the 7.6 L. 197 to 7.5.

*L.260: remove "(not NCP)"*

We removed "(not NCP)".

*L.334: day:night period in September should be around 12:12*

See response to RC1: The times are given as a broad indication. Precisely, the sunrise to sunset period was:

- 14:25 to 14:29 hours on 28-29 April 2021
- 10:07 to 10:17 hours on 14-17 February 2020

- 9:50 to 10:30 hours on 9-19 February 2021
- 13:20 to 12:45 hours on 2-11 September 2020
- 13:07 to 12:56 hours on 6-9 September 2021

The time before sunrise and sunset in the summer period we investigated is thus closer to 13 hours. With twilight times being particularly long in September (around 1 hour of light before sunrise/after sunset), we considered that 14:10 for September was a good approximation of "day:night", when considering "night" as "no light".

L.394: I suggest writing "$CO_2$ addition increased $O_2$-derived $NCP_{lm}$ by 20% on average over all seasons". Same L.401

Thank you for this suggestion. The text now reads: "$CO_2$ addition increased $O_2$-derived $NCP_{lm}$ by 20% on average over all seasons, from $10 \pm 7$ mmol $O_2$ m-2 hr -1 in "present" conditions to $12 \pm 9$ mmol $O_2$ m-2 hr -1 (p = 0.0015)." (L 394) and "As for $O_2$ -derived NCP lm $CO_2$ addition increased DIC-derived NCP lm by 20 % on average over all seasons (p < 0.001, Fig. 4A)." (L 401).

L.440: "$\approx 10$ mg $O_2$ $L^{-1}$" please give the exact value.

We changed the sentence to "After five hours of emersion, $O_2$ concentration had decreased by half (from $10.1 \pm 1.5$ mg L -1 to $4.9 \pm 3.3$ mg $O_2$ L-1)"

L.505: It does not seem reasonable to me to extrapolate the CCB calculation by considering the day:night duration. First of all, the emersion cycles are ~6-7h (2 cycles per day) and tide pools are therefore not exposed to day:night 10:14h in winter during emersion. In order to extrapolate, one should also consider the CCB during the immersion cycles, and then calculate the budget with a day:night photoperiod of 10:14. I suggest removing this sentence.

This was also pointed out by RC1 and we removed the sentence entirely.

L.521: Tide pool communities are subjected to extremely variable environmental conditions for a few hours during emersion. Immersion cycles play an important role in the ability of species to colonize these upper-shore environments by renewing seawater in tide pools. Therefore, in a future scenario of ocean acidification, tide pools would be subjected to low pH conditions during emersion at night, but also during immersion cycles. This is likely to affect the ability of these communities to cope with ocean acidification, as they would be exposed to low pH conditions for longer periods of time. A few lines should be added about this in the discussion.

This is a very valid point that we had not discussed in this manuscript. We added a paragraph on this subject (line 507):

« In the current study, we only consider the tidal pools as closed (emersed) systems. However, in an acidifying ocean, tidal pool communities will also be affected by lowered pH during immersion, resulting in longer exposure to low pH. More realistic budgets would thus need to integrate these immersion periods, which might have possible additive negative effects on calcification (see e.g., Legrand et al., 2018 for tidal assemblage experiments on net production/respiration)."

Legrand, E., Riera, P., Bohner, O., Coudret, J., Schlicklin, F., Derrien, M., & Martin, S. (2018). Impact of ocean acidification and warming on the productivity of a rock pool community. *Marine Environmental Research*, *136*, 78–88. https://doi.org/10.1016/j.marenvres.2018.02.010

L.536: Confusing. I suggest writing "The community's net primary production increased by 20% on average across all seasons, and was particularly visible in summer (+ 35%)"

We replaced "This increase in the community's net primary production by 20% was particularly visible in summer (+ 35%), at higher temperatures/metabolic rates." by the reviewer's suggestion: *"The community's net primary production increased by 20% on average across all seasons, which was particularly visible in summer (+ 35%)".*

**Technical corrections:**

L.38: Subscript for $CO_2$

Thank you for pointing this out, this was corrected.

L.205: add coma after "0 to 8 mg $L^{-1}$"

Thank you for pointing this out, this was corrected.

---

## Author Comment (AC2)

**Answers to reviewers, Dorey et al., BGD (bg-2023-79).**

We would like to thank the reviewers for their constructive criticism and the time they spent on careful proofing of our submitted manuscript. Answer to the reviewers can be found below in blue: The lines mentioned refer to the original version.

**RC1: 'Comment on bg-2023-79', Zvi Steiner, 29 Jun 2023 reply**

Acidification of the ocean due to the ongoing anthropogenic CO2 emissions is affecting calcification and photosynthesis in the ocean. However, the effect of acidification on the community level is complex as different organisms might respond differently to this issue. Dorey et al. studied the effect of ocean acidification in natural rock pools, which is an interesting approach as the rock pools are isolated from the ocean during low tide forming a perfect mesocosm experiment with a natural benthic community. Conditions in the rock pools naturally vary greatly between day and night and between seasons, hence the community is expected to be fairly resilient to acidification, yet the experiments show a clear effect of the interventions.

*My main concern about the paper is that it lacks a proper inorganic control. The assumption of the authors is that there is no equilibration of* CO2 *added to the pools with atmospheric* CO2. *This is a fair assumption in the open ocean, as it takes* CO2 *several years to equilibrate with the atmospheric pressure but* I'm not sure *it holds in the experimental setup because of the large deviation of the* pCO2 *in the treated pools from atmospheric values, fluctuations in pool temperature during the day (particularly in summer), and the large surface area and limited volume of the pools that are also exposed to the elements. Therefore, I think it is important to add a control experiment that will test if there is a change in seawater* pCO2 *after a* CO2 *enrichment at the study site in summer conditions. The control experiment can be done in a simple tub with similar dimensions to one of the pools, that is initially filled with artificial or filtered seawater and treated similar to the* CO2 *enrichment experiment. It might also be useful to bubble* N2 *into a second identical tub to stimulate low* CO2 *and* O2 *conditions and test the rate of penetration of these gases during a few hours.*

Authors: We thank the reviewer for pointing this study limitation out and while we are unable to do the proposed additional experiment for the moment, we will present below why we consider the  $CO_2$  air-sea diffusive transport to be negligible using the theoretical fluxes of  $CO_2$  in a 0.5 m2 pool treated with additional  $CO_2$  (+1500 µatm).

First, we need to explain why we do not consider the effects of wind and waves in the air-sea gas exchanges in our tidal pool setting.

The effect of wind is very minimal on the pools, as the pools are well-protected from the wind by higher rocks on the floor, in a boundary layer where surface friction is high. Furthermore, with its eastern orientation, the Beach of Bloscon is generally sheltered from the usual western and south-western winds (see for example the figure below presenting the wind direction and figure 2.3 in nearby Brest in the thesis here: https://www.researchgate.net/publication/286473561\_Dynamique\_et\_echanges\_sed imentaires\_en\_rade\_de\_Brest\_impactes\_par\_I%27invasion\_de\_crepidules/figures?lo =1, or in a review by the national meteorological institute *Météo France*, here: https://donneespubliques.meteofrance.fr/donnees\_libres/bulletins/BCMR/BCMR\_05\_ 201401.pdf).

There were no movement/waves in the pools, nor was there any visible breeze waves. The effect of surface wave on gas transfer velocity is generally neglected in small lakes and even though, it is discussed for larger lakes such as Lake Geneva (e.g., Perolo et al. in ESD, 12, 1169–1189, 202, https://esd.copernicus.org/articles/12/1169/2021/), it can reasonably be neglected in our small tidal pools.

There is no vertical or horizontal transport of  $CO_2$  due to convection in those small pools since we are only interested in them when they are closed at low tide.

We can thus consider the flux of  $CO_2$  at the surface of the tidal pools is only due to net diffusive transport:

**$F_{CO2} = k \ a \ \Delta pCO_2$**

Where **F** is the CO2 flux (µmol cm-2 h-1), **a** is the CO2 solubility coefficient (umol cm-3 atm-1), **k** is the gas transfer velocity (cm h-1), and **\Delta pCO\_2** is the gradient of partial CO2 pressure ( $pCO_2$ ) between the water and the air (see example references for this equation in Perolo et al., 2021, in ESD, 12, 1169–1189, 202, https://esd.copernicus.org/articles/12/1169/2021/). We also consider that there is no need to correct for atmosphere (pressure = 1atm) as we are at sea level at Paris latitude.

- For the solubility coefficient of  $CO_2$ , a table depending on water temperature and salinity can be found in Weiss (1974) Mar Chem, 2 (3), 203-215

(https://www.sciencedirect.com/science/article/pii/0304420374900152): at a temperature 20°C and a salinity of 35,  $\mathbf{a} = 3.32 \ \mathbf{10}^{-2} \ \mathbf{mol} \ \mathbf{L}^{-1} \ \mathbf{atm}^{-1}$ .

- For the  $\Delta pCO_2$  we take the average treatment of +1500 µatm compared to a "control" pool (i.e., a pool equilibrated to the atmosphere where  $\Delta pCO_2 = 1$ ).  $\Delta pCO_2 = 1.5 \ 10^{-6} \ \text{atm.}$
- For the gas transfer velocity, we used the data and personal communication from David Ho (Ho et al., 2018 *On factors influencing air-water gas exchange in emergent wetlands.* Journal of Geophysical Research: B 123, 178–192, https://agupubs.onlinelibrary.wiley.com/doi/epdf/10.1002/2017JG00429). David Ho that told us that, he had found an estimation of  $k = 1 \text{ cm h}^{-1}$  the method in small artificial pools. The result  $k = 1 \text{ cm h}^{-1}$  is also what is presented in baseline field conditions in the Everglades the paper Ho et al. (2018) cited above.

Therefore:

 $F_{CO2} = k a \Delta pCO_2$

 $F_{CO2} = 1 \text{ cm } h^{-1} \times 3.3 \ 10^{-2} \text{ mol } L^{-1} \text{ atm}^{-1} \times 1.5 \ 10^{-6} \text{ atm}$   $F_{CO2} = 1 \text{ cm } h^{-1} \times 3.3 \ 10^{-5} \text{ mol } \text{ cm}^{-3} \text{ atm}^{-1} \times 1.5 \ 10^{-6} \text{ atm}$   $F_{CO2} = 4.95 \ 10^{-11} \text{ mol } \text{ cm}^{-2} \text{ h}^{-1}$   $F_{CO2} = 0.495 \ \mu\text{mol } \text{m}^{-2} \text{ h}^{-1}$

In our study, typical pool size was 0.5 m2, or 5000 cm2. Thus, the seawater of a treatment pool (+1500 µatm) "loose" an approximative **4.95 10-11 mol cm-2 h-1 x 5000 cm2** = 2.5 10-7 mol h-1 to the atmosphere, or **0.25 µmol per hour due to net diffusive transport**.

In the present study (see Figures 3), the biological rates outweigh by a factor 100 this net diffusive transport. For instance, calcification rates (a rate calculated using total alkalinity, not DIC/CO2 concentrations) range from -2 to +4 mmol m-2 h-1, i.e., a change by -1 to +2 mmol per hour in a 0.5 m2 pool. Compared to calcification, the change in carbon due air-sea exchanges is thus less than 0.1%.

To put this result - a change by 0.25  $\mu$ mol per hour due to net diffusive transport - further in perspective, data we obtained using individual incubations conducted in the same pools conditions in summer, that gastropods (*Gibbula pennanti*) respire at rates of 10  $\mu$ mol h-1 per individual.

Even when considering higher values for k and a, it seems unlikely that the air-sea flux would play a major role in the observed changes. For instance, in winter, water at 5°C would only multiply the gas solubility by 2 ( $a = 5.4 \ 10^{-2} \ mol \ L^{-1} \ atm^{-1}$ ). In in natural field conditions of the Everglades (Ho et al. 2018), the maximum values found were  $k = 3 \ cm \ h^{-1}$ . In Geneva's Lake, stronger winds (> 5 m s-1) led to k values around

30-40 cm h-1 (Perolo et al., 2021). However, even these gas transfer velocity levels which would not increase gas exchanges to a considerable level:

**$F_{CO2} = 40 \text{ cm h}^{-1} \text{ x 5 10}^{-2} \text{ mol L}^{-1} \text{ atm}^{-1} \text{ x 1.5 10}^{-6} \text{ atm}$ = 2.25 10-9 mol cm-2 h-1**

We thus conclude that the air-sea exchanges were negligible, and that they did not affect the treatments. We also added a sentence in the material and methods to reflect this "The air-sea fluxes due to net diffusive transport are considered to be negligible (see *answer to reviewers*)."

Out of curiosity, and depending on our abilities, we may however conduct an experiment with artificial seawater in early September in Roscoff.

**RC1**: The discussion about nocturnal dissolution should include more references about supersaturated CaCO3 dissolution in seawater – you can start with Milliman et al., DSR, 1999 and Subhas et al., GBC, 2022, and references therein.

Authors: We thank the reviewer for pointing out to us those two articles:

Milliman, J. D., Troy, P. J., Balch, W. M., Adams, A. K., Li, Y.H., and Mackenzie, F. T.: Biologically mediated dissolution of calcium carbonate above the chemical lysocline? Deep-Sea Research I, 46, 1653–1669, 1999. https://www.sciencedirect.com/science/article/abs/pii/S0967063799000345

Subhas, Zvi Steiner et al. 2022. Shallow Calcium Carbonate Cycling in the North Pacific Ocean https://agupubs.onlinelibrary.wiley.com/doi/abs/10.1029/2022GB007388

However, after careful reading, we did not know where they would fit in our discussion as they refer to plankton and the open ocean.

**Other comments:**

**Line 283: Data is plural.**

We changed "data was limited to the first three hours of emersion as high O2 concentrations" to "data were limited to the first three hours of emersion as high O2 concentrations".

**Line 331: Is the total alkalinity concentration salinity normalised?**

The total alkalinity presented here was not salinity normalized.

**Line 334: day:night period should be close to 12:12 in September.**

The times are given as a broad indication. Precisely, the sunrise to sunset period was:

- 14:25 to 14:29 hours on 28-29 April 2021
- 10:07 to 10:17 hours on 14-17 February 2020
- 9:50 to 10:30 hours on 9-19 February 2021
- 13:20 to 12:45 hours on 2-11 September 2020
- 13:07 to 12:56 hours on 6-9 September 2021

The time before sunrise and sunset in the summer period we investigated is thus closer to 13 hours. With twilight times being particularly long in September (around 1 hour of light before sunrise/after sunset), we considered that 14:10 for September was a good approximation of "day:night", when considering "night" as "no light".

**Lines 404-5: Not clear, do you refer to the absolute productivity or to the change in productivity due to the interventions?**

Agreed, this sentence was unclear. We transformed "Compared to pool A, productivity was significantly lower in pools B and E and significantly higher in pools C and D (p < 0.003)." by "NCPIm was significantly lower in pools B and E than in pool A, and significantly higher in pools C and D (p